# The mechanism of ribosomal recruitment during translation initiation on the Type 2 encephalomyocarditis virus IRES

Sayan Bhattacharjee [ID] [1,3], Irina S Abaeva [ID] [2,3], Zuben P Brown [1,3], Yani Arhab [ID] [2,3], Hengameh Fallah [1], Christopher U T Hellen [ID] [2 ✉], Joachim Frank [ID] [1 ✉] & Tatyana V Pestova [ID] [2 ✉]

## Abstract

The encephalomyocarditis virus (EMCV) internal ribosomal entry side (IRES) and other Type 2 IRESs favor translation of the viral genome during infection. The domains H-L of these IRESs specifically interact with the cellular translation initiation factors eIF4G/eIF4A through their essential JK domain. However, the JK domain is not sufficient for IRES activity, which also strictly requires the preceding domain I of unknown function. To identify interactions that drive ribosomal attachment to eIF4G/eIF4A-bound Type 2 IRESs, we determined the cryo-EM structure of 48S initiation complexes formed on the EMCV IRES. The apical cloverleaf of domain I contacts ribosomal proteins uS13 and uS19 via its subdomain Id, whereas the essential GNRA tetraloop in subdomain Ic interacts with the TψC domain of initiator tRNA. The IRES-tRNA interaction also provides a mechanism for release of the IRES after eIF2 is replaced by eIF5B during subunit joining to allow attachment of 60S subunits. Functional assays supported the exceptional role of these interactions for initiation on this IRES. The strong conservation of the apex of domain I amongst Type 2 IRESs suggests that the reported interactions provide a common general mechanism of ribosomal attachment on them all.

**Keywords** IRES; Translation initiation; GNRA tetraloop; Cryo-EM; Picornavirus
**Subject Categories** Structural Biology; Translation & Protein Quality

## Introduction

Initiation on most cellular mRNAs occurs by the 5'end-dependent ribosomal scanning mechanism and involves multiple eukaryotic initiation factors (eIFs) (Jackson et al, 2010). A 40S ribosomal subunit binds an eIF2•GTP/Met-tRNA$_i^{Met}$ ternary complex, eIFs 3, 1 and 1A to form a 43S preinitiation complex (PIC). Attachment of 43S PICs to the capped 5'-end of mRNA is mediated by eIF4F (comprising the cap-binding protein eIF4E, the RNA helicase eIF4A, and the scaffold protein eIF4G which also binds to eIF3), eIF4A and eIF4B. eIFs 4A/4B/4F unwind the cap-proximal region of mRNA allowing attachment of 43S PICs that is also facilitated by the eIF4G-eIF3 interaction. 43S PICs subsequently scan to the first AUG in a favorable context, where they form 48S initiation complexes (ICs) with codon-anticodon base-pairing in the ribosomal P site. eIF1, in cooperation with eIF1A, monitors the fidelity of initiation. Start codon recognition leads to eviction of eIF1, eIF5-induced hydrolysis of eIF2-bound GTP and Pi release, committing the 40S subunit to the initiation codon. Subsequent joining of a 60S ribosomal subunit and release of eIFs are mediated by eIF5B, which reorients the acceptor end of Met-tRNA$_i^{Met}$ to allow subunit joining (Brown et al, 2022; Lapointe et al, 2022) and then hydrolyzes GTP and dissociates from assembled 80S ribosomes.

Viruses have evolved diverse mechanisms to sustain translation of their mRNAs in infected cells after innate immune responses have been activated. One approach employs an internal ribosomal entry site (IRES), which is a *cis*-acting RNA element that promotes end-independent ribosomal recruitment to an internal location in mRNA (Mailliot and Martin, 2018). There are four principal classes of IRES: Type 1 (e.g., poliovirus), Type 2 (e.g., encephalomyocarditis virus (EMCV)), Type 4 (e.g., hepatitis C virus (HCV)) and Type 6 (e.g., cricket paralysis virus (CrPV) and Halastavi árva virus (HalV)) (nomenclature according to Arhab et al (2020)). These classes employ distinct mechanisms for ribosome recruitment that are nevertheless all based on specific non-canonical interactions with canonical components of the translation apparatus and require only a subset of canonical eIFs. Thus, Type 4 IRESs bypass the requirement for eIFs 4A, 4B, 4F and eIF3 by binding directly to the platform of the 40S subunit, usurping the eIF3-binding site (Pestova et al, 1998b; Hashem et al, 2013; Brown et al, 2022), whereas type 6 IRESs bind to either the ribosomal A site (CrPV (Wilson et al, 2000; Schüler et al, 2006)) or to the P site (HalV (Abaeva et al, 2020)) and do not require any eIFs or initiator tRNA. On Type 4 IRESs, Met-tRNA$_i^{Met}$ can be recruited not only by eIF2 but also by eIF5B or even eIF2D (Pestova et al, 2008; Terenin et al, 2008; Skabkin et al, 2010).

[1]Department of Biochemistry and Molecular Biophysics, Columbia University, New York, NY 10032, USA. [2]Department of Cell Biology, SUNY Downstate Health Sciences University, Brooklyn, NY 11203, USA. [3]These authors contributed equally: Sayan Bhattacharjee, Irina S Abaeva, Zuben P Brown, Yani Arhab. ✉E-mail: christopher.hellen@downstate.edu; jf2192@cumc.columbia.edu; tatyana.pestova@downstate.edu

Importantly, cryo-EM studies have been indispensable in gaining molecular insights into the mechanisms of initiation on Type 4 and Type 6 IRESs (Schüler et al, 2006; Hashem et al, 2013; Fernández et al, 2014; Muhs et al, 2015; Quade et al, 2015; Yamamoto et al, 2015; Murray et al, 2016; Abeyrathne et al, 2016; Pisareva et al, 2018; Acosta-Reyes et al, 2019; Abaeva et al, 2020; Brown et al, 2022).

Type 2 IRESs were initially discovered in members of *Picornaviridae* (Jang et al, 1988; Kühn et al, 1990) and were recently also identified in *Caliciviridae* (Arhab et al, 2022, 2024). They are ~450-nt-long and have five principal domains (H-L), with a Yn-Xm-AUG motif at their 3′-border in which a Yn pyrimidine tract (n = 8–10 nt) is separated by a spacer (m = 18–20 nt) from an AUG triplet that acts as the initiation codon (shown for the EMCV IRES in Fig. EV1). In the EMCV IRES, the primary initiation site is $AUG_{834}$, which is a part of the Yn-Xm-AUG motif, whereas the preceding $AUG_{826}$ is only weakly utilized during in vitro reconstitution (e.g., Pestova et al, 1998a). 48S complex formation can also occur at a lower level at $AUG_{846}$, in-frame with $AUG_{834}$, during in vitro reconstitution (Abaeva et al, 2023) as well as during in vitro translation in rabbit reticulocyte lysate (RRL) in the presence of GMPPNP (Dmitriev et al, 2003). The fact that in the presence of GTP, 80S complexes form in RRL almost exclusively at $AUG_{834}$ (Dmitriev et al, 2003) suggests that 40S subunits can slide downstream during prolonged incubation if the next stage in translation is blocked, suggesting that $AUG_{846}$ can act as a safety trap in case of a delay in subunit joining. In vitro reconstitution has shown that initiation on Type 2 IRESs requires eIF2, eIF3, eIF4A and the central domain of eIF4G, that it is stimulated by eIF4B and the pyrimidine tract-binding protein PTB (or its neuronal paralogue nPTB), an IRES *trans*-acting factor (ITAF), and that eIF1 and eIF1A synergistically enhance the fidelity of initiation codon selection (Pestova et al, 1996a, 1996b, 1998a; Pilipenko et al, 2000; Andreev et al, 2007).

Each domain in type 2 IRESs has a conserved structure and contains highly conserved sequence motifs, mostly at peripheral locations, that are critical for function. PTB binds to oligopyrimidine sequences in different domains and is thought to stabilize the IRES in an active conformation (Kafasla et al, 2009; Yu et al, 2011). The only known essential interaction with canonical components of the translation apparatus for Type 2 IRESs is that of their JK domains with eIF4G's central domain (Pestova et al, 1996a, 1996b; Kolupaeva et al, 1998; López de Quinto and Martínez-Salas, 2000; Clark et al, 2003; Imai et al, 2016, 2023) (Fig. EV1). eIF4G binds to highly conserved elements of the J-K domain in a manner that is enhanced by eIF4A (Lomakin et al, 2000; Imai et al, 2016, 2023), and together, they induce conformational changes at the IRES's 3' border, in the region that enters the mRNA binding cleft of the 40S subunit (Kolupaeva et al, 2003). The JK domain is essential, but it is not sufficient for IRES activity, which also requires domain I (Kühn et al, 1990; Evstafieva et al, 1991; Duke et al, 1992). The function of domain I remains obscure. It consists of a long, irregular stem and an apical double cross structure with conserved loops, including a GNRA tetraloop. The apical region of domain I in the foot-and-mouth disease virus IRES was reported to undergo conformational changes upon incubation with ribosomal subunits (Lozano et al, 2018), but the functional significance of these changes was not known. GNRA tetraloops adopt a specific U-turn structure which is strongly stabilized by networks of hydrogen bonding and base-stacking within the loop (Heus and Pardi, 1991; Jucker et al, 1996; Wedekind and McKay, 1998). This structure is characterized by a sharp turn of the phosphate backbone between G and the second nucleotide and the last three bases being sequentially stacked on each other. This structure exposes the Watson-Crick base-pairing edge of these three bases, enabling GNRA tetraloops to engage in long-range tertiary interactions (Fiore and Nesbitt, 2013). GNRA tetraloops are consequently often involved in interactions with other RNA elements (e.g., in ribosomal RNAs) or proteins, playing a key role in RNA folding and function. In Type 2 IRESs, GNRA loops are also critical for the IRES activity, but their exact function and potential binding partners remain unknown (Roberts and Belsham, 1997; López de Quinto and Martínez-Salas, 1997; Robertson et al, 1999; Nateri et al, 2000; Fernández-Miragall and Martínez-Salas, 2003; Fernández et al, 2011).

The facts that the JK domain is not sufficient for EMCV IRES function and that initiation on it does not depend on eIF4G's eIF3-binding domain (Lomakin et al, 2000; Sweeney et al, 2014) raise the question of which interactions are responsible for recruiting an eIF4G/eIF4A-bound Type 2 IRES to a 43S PIC. To resolve this question, which is of paramount importance for this mechanism of initiation, we determined the cryo-EM structure of 48S complexes assembled on the EMCV IRES.

## Results

### The overall cryo-EM structure of the 48S complex assembled on the EMCV IRES

To identify the interactions that are responsible for the recruitment of Type 2 IRESs to 43S preinitiation complexes, we determined the cryo-EM structure of the 48S initiation complex assembled on the EMCV IRES. 48S complexes were reconstituted in vitro on EMCV IRES-containing mRNA (nt 315–862) using individual 40S ribosomal subunits, Met-tRNA$_i^{Met}$, initiation factors eIF2, eIF3, eIF4A, eIF4B, eIF4F, eIF1, eIF1A, as well as the EMCV-specific ITAF nPTB (Fig. EV2A). In vitro reconstitution done using purified components ensures the assembly of complexes of known composition. Accurate assembly of 48S complexes on the IRES was verified by primer extension inhibition, which involves extension by reverse transcriptase of a primer base-paired to the mRNA. mRNA-bound ribosomes arrest primer extension, yielding toe-prints at their leading edge that can be located on sequencing gels. 48S complexes yield stops 15, 16 and 17 nt downstream of the first nucleotide ($^+$1) of the initiation codon. To increase homogeneity, the weakly utilized $AUG_{826}$ (Pestova et al, 1998a) was mutated to eliminate 48S complex formation on this codon (Fig. EV2B). To avoid potential cryo-EM artifacts, chemical cross-linking was not applied following 48S complex assembly.

Grid preparation, data collection and image processing are described in Methods. Cryo-EM grids were imaged at 300 kV producing high-resolution micrographs with easily identifiable 40S ribosomal particles (Appendix Table S1). From initial 3D classification (see Methods and Fig. EV3A) four classes of interest were selected: Class I (134,626 particles), Class II (98,748 particles), Class III (89,242 particles), and Class IV (72,526 particles). All four classes exhibited densities corresponding to the 40S ribosomal

subunit, eIF1A and the IRES bound to the inter-subunit face of the 40S head. Classes II–IV also contained density for eIF1, whereas not fully resolved Class I contained additional density for eIF2, Met-tRNA$_i^{Met}$, and blurry density for eIF3. Subsequent focused classification on the Class I population revealed a refined class, designated Class Ia, containing 59,153 particles with well-resolved density for the 40S, eIF2, eIF1A, Met-tRNA$_i^{Met}$, eIF3 and the IRES. Additionally, Classes II–IV that could correspond to initial binding of the IRES to 40S subunits differed from one another in the extent of the rotation of the 40S subunit head and in repositioning of the IRES (Fig. EV3B), which might represent conformational changes that would gradually open the P site and reduce potential steric clashes for spatial accommodation of Met-tRNA$_i^{Met}$, eIF2α and the IRES. The smaller particle population observed in Class Ia compared to the other classes likely reflects dissociation of eIF3 from 48S complexes during grid preparation because 48S complexes did not undergo chemical cross-linking. The lack of cross-linking could also account for the absence of eIF4F/eIF4A in our structure (Brito Querido et al, 2020). Subsequent refinement of density maps from Classes Ia–IV yielded structures with an approximate global resolution of 3.2 Å according to FSC curves (see Methods; Fig. EV3 and Appendix Fig. S1).

In the final model (Fig. 1A–C), 48S complexes showed density corresponding to canonical components of initiation complexes i.e., the 40S subunit, P-site Met-tRNA$_i^{Met}$ base-paired with the initiation codon and bound to the eIF2γ subunit, E-site eIF2α subunit, A-site eIF1A, the five-lobed PCI/MPN core of eIF3 bound to its conventional site at the solvent side of the 40S subunit, as well as an additional density for the IRES (red) bound to the head of the 40S subunit and interacting with ribosomal proteins uS13 and uS19 (blue and green, respectively) as well as with the P-site Met-tRNA$_i^{Met}$ (magenta), and also extending away from the ribosome head.

Due to the high resolution of the IRES, we could identify contiguous strands of mRNA and classify nucleotides within those strands as either purines (R) or pyrimidines (Y) based on the size of their density. To determine which region of the IRES was bound to the 40S subunit we chose the highest-resolution portion, a loop bound in the pocket formed between alpha helix 2–3 of ribosomal protein uS13 and the beta sheet of uS19 (Fig. EV4). This region of the IRES consists of 13 nucleotides beginning with a clearly identifiable purine, followed by a pyrimidine, two purines, seven nucleotides in a loop and two pyrimidines. Using the presumptive identity of these nucleotides we searched the IRES for candidate regions beginning with the smallest identifiable sequence (viz. RNRR) and found 39 possible locations. The IRES density clearly showed a 7-nt-long loop linking the final two purines of our search term (Rn**RR**) to two facing nucleotides. We therefore eliminated any region that could not accommodate such a loop from further consideration and were left with two candidates, in domain H ($^{414}$AGGG$_{417}$) and in subdomain Id ($^{562}$GCGG$_{565}$). Examination of the IRES density suggests that an additional pyrimidine is present at either end of the search term (**Y**RNRR**Y**) and by examining each separately (e.g., RNRRY and YRNRR) we eliminated domain H as a possibility due to the length of the loop. We exhaustively increased the length of the search term as well as the direction (e.g., 5′ to 3′ or 3′ to 5′) (Appendix Table S2) and in all cases the sequence in subdomain Id was identified as the only feasible candidate. Although the density of the loop suggested an unambiguous length

of seven nucleotides connecting the two purines ($^{564}$GG$_{565}$) to their base-pairing pyrimidines ($^{573}$CC$_{574}$), we also varied the length during our search to minimize bias and found no possibilities that conformed with the predicted secondary structure of the IRES. Full details of the parameters that we used to identify possible candidate sequences are given in the Methods. Once we had determined the identity of the $_{561}$UGCGGCCAAAAGCC$_{574}$ region, we were able to build subdomains Ib, Ic, Id, and Ie that span 83 nucleotides in the apex of domain I ($_{518}$CC..GG$_{600}$) accounting for 17% of the IRES (Figs. 1D–G and EV1), and that interact with Met-tRNA$_i^{Met}$, uS13 and uS19 (Fig. 1H–K).

To verify the assignment of cryo-EM density to the apical region of domain I and to confirm its identical position in 48S complexes in solution, we performed directed hydroxyl radical cleavage of the IRES from the surface of eIF1A in in vitro assembled 48S complexes and also identified the regions of the IRES that are protected from RNase T1 digestion in initiation complexes. We observed hydroxyl radical cleavage at the junction of Id and Ie subdomains (CCA$_{573-575}$) from two positions in the C-terminal tail of eIF1A (D$_{132}$ and D$_{137}$) (Appendix Fig. S2A,B). The modeled position of these IRES nucleotides is in very close proximity to 18S rRNA nucleotides that are cleaved from the same positions of eIF1A in 43S preinitiation complexes (Yu et al, 2009) (Appendix Fig. S2C), confirming the assignment. Protection from RNaseT1 cleavage of G$_{549}$ in subdomain Ic and of G$_{585}$ in subdomain Ie was observed in reaction mixtures for 48S complex formation (Appendix Fig. S3A, left panel; Appendix Fig. S3C) and in assembled, sucrose density gradient purified 48S complexes (Appendix Fig. S3A, right panel; Appendix Fig. S3C), which is also consistent with the cryo-EM structure. In addition, we observed protection of G$_{474}$ in the central stem of domain I, an element that was not seen by cryo-EM. Protection of G$_{775}$ and G$_{761}$ in domain JK (Appendix Fig. S3B,C) corresponded to the previously reported specific binding of eIF4G/eIF4A (Pestova et al, 1996b; Kolupaeva et al, 1998).

## Specific interactions of the EMCV IRES with the proteins uS13 and uS19 and initiator tRNA

Detailed analysis of the model revealed that the apical region of IRES domain I adopts a cloverleaf conformation that allows the Ih2 helix and the A-rich Id loop to interact with ribosomal proteins uS13 and uS19, respectively (Fig. 2). Specifically, the α-helix of uS13, composed of charged and polar residues E112, R108, and N105, forms a platform that facilitates docking of the IRES Ih2 helix via its G530 and U561 residues (Fig. 2A–F). Further interaction of Q104 and N103 of uS19 with the Id loop helps to support this docking of the IRES to the 40S subunit head (Fig. 2G–L).

Strikingly, the Ic GNRA loop (GCGA$_{547-550}$), which is essential but whose mechanism of action is unknown, is positioned to interact directly with the initiator tRNA TψC domain (Fig. 3A–G). Comparison of the P-site tRNA model in canonical cap-dependent 48S ICs (e.g., Brito Querido et al, 2024) and in the EMCV IRES 48S IC revealed that while binding to the P-site codon the tRNA acceptor domain shifts by 9.7 Å from the associated eIF2 to interact with the EMCV IRES (Fig. 3B). In the IRES-tRNA interface, bases A550, G549 and C548 in the GNRA loop are connected either via electrostatics or via van der Waals force with G48 and C65 in

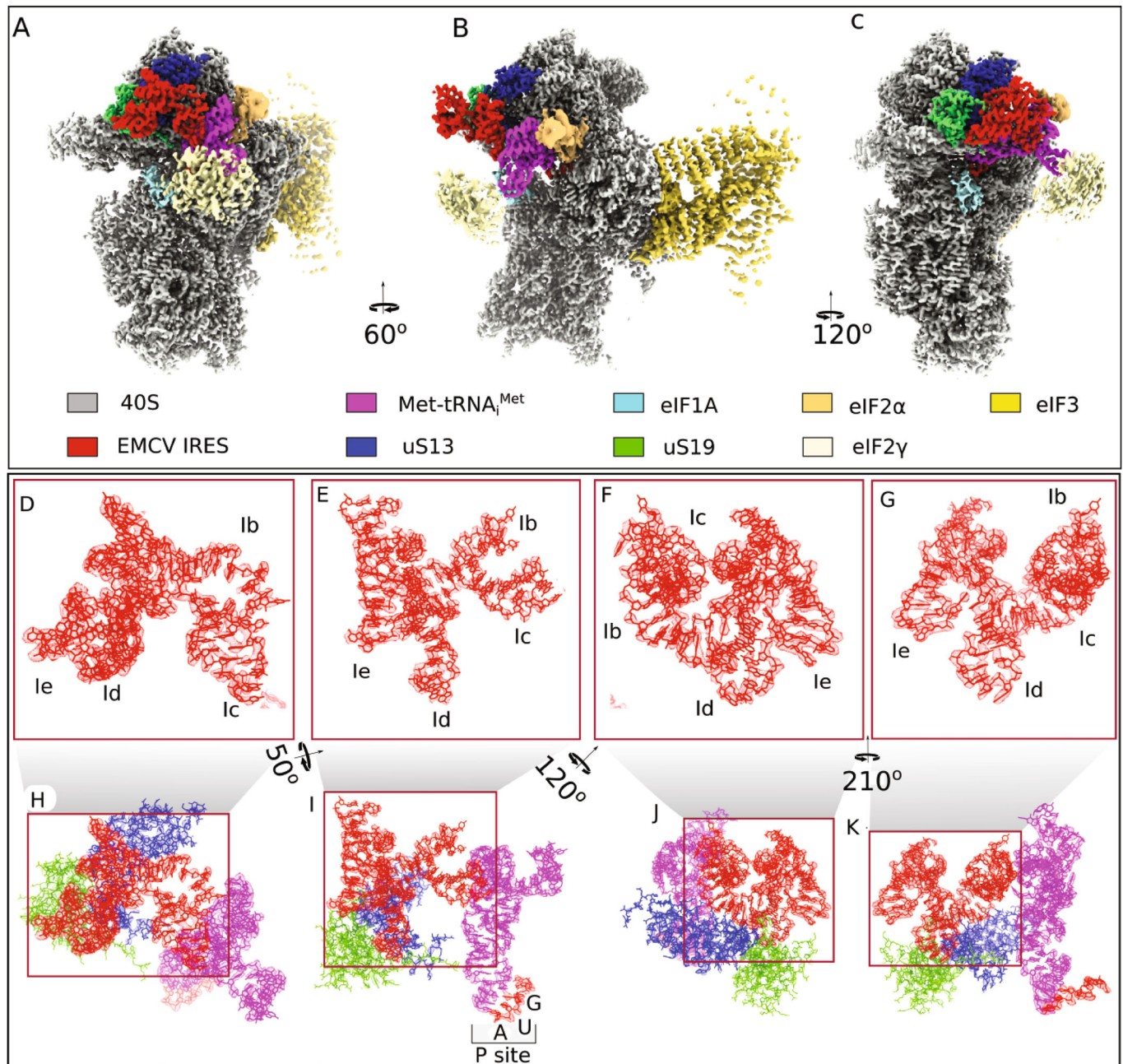

**Figure 1. Translation initiation complex on the EMCV IRES.**

(**A–C**) Cryo-EM map of the 48S initiation complex assembled on the EMCV IRES, shown in different orientations. The 40S ribosomal subunit, ribosomal proteins uS13 and uS19, associated initiation factors, Met-tRNA$_i^{Met}$ and EMCV IRES are shown in distinct colors and are labeled accordingly. (**D–G**) Zoomed-in views of the EMCV IRES density map (rendered at 70% surface transparency) superimposed with the atomic model, displayed in different orientations. The individual domains of the IRES are indicated in these views. (**H–K**) Zoomed-in views of the cryo-EM density map, with superimposed atomic models, showing the key interactions between the EMCV IRES (red) and initiator tRNA (magenta), uS13 (blue) and uS19 (green). The densities of the different components are color-coded using the ChimeraX surface color tool (Meng et al, 2023). Panel (**I**) also shows interaction between the start codon (AUG) of the mRNA and the anti-codon of the initiator tRNA at the P site of the 40S subunit.

initiator tRNA (Fig. 3D–G). The IRES-Met-tRNA$_i^{Met}$ interaction is further stabilized by the additional interaction of G546 and A550 in the IRES and U46 in tRNA (Fig. 3H,I), where the Met-tRNA$_i^{Met}$ variable loop containing U46 rotates to interact with the IRES.

The IRES-induced displacement of the tRNA acceptor arm from its position associated with eIF2β (Appendix Fig. S4) and rotation

of the 40S subunit head also leads to destabilization of the local interaction network compromising the rigidity of the uncross-linked complex further and therefore contributing to the absence of well-resolved density for eIF2β in our reconstruction.

Comparison with available structures of GNRA tetraloops revealed that the GNRA tetraloop of the IRES has a similar

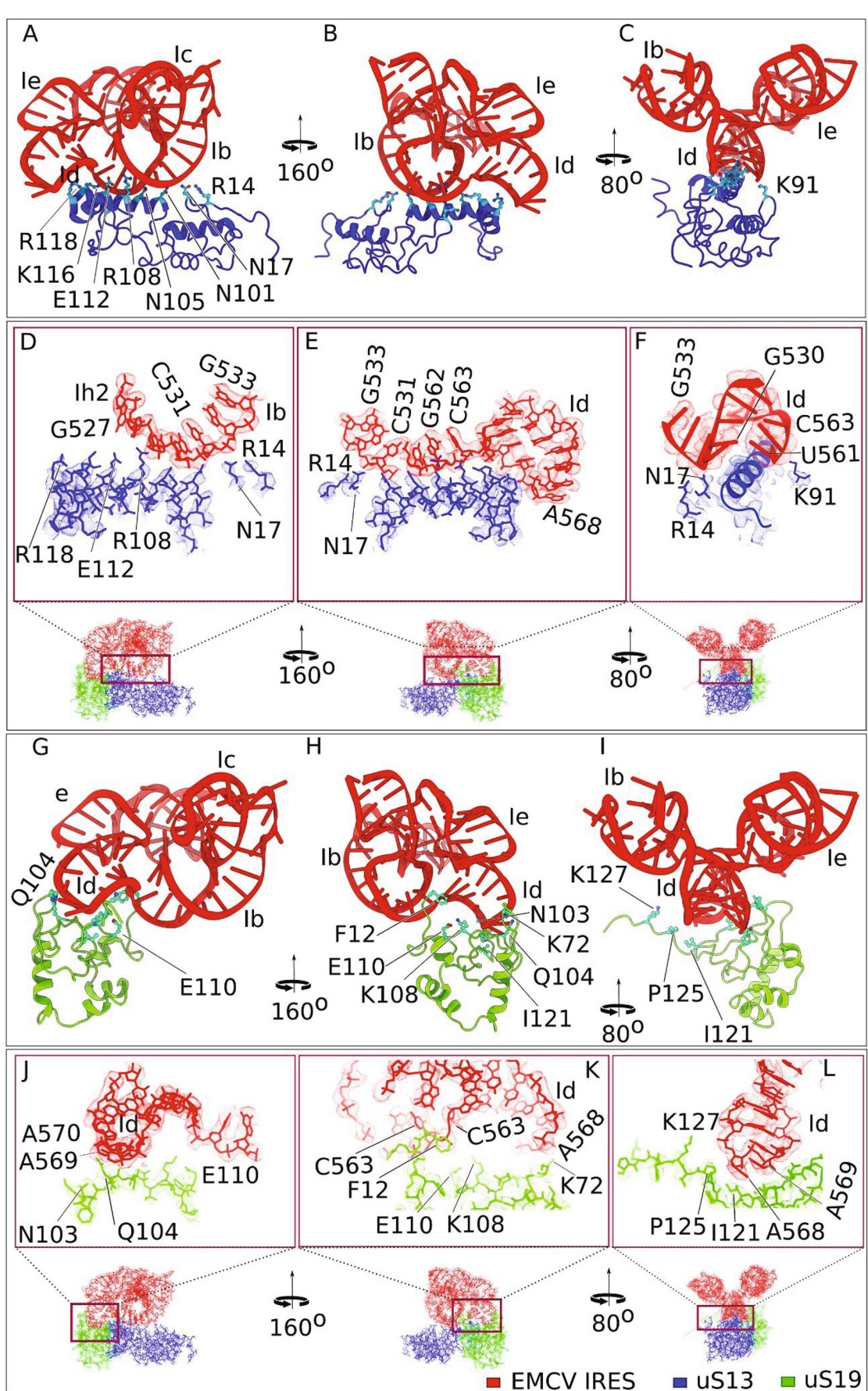

**Figure 2. Specific interactions of EMCV IRES domain I with the head of 40S ribosomal subunit.**

(A–C) Zoomed-in views of the interaction between protein uS13 (blue) of the 40S subunit and the EMCV IRES, as shown by atomic models in three orientations, with interacting residues labeled. (D–F) Close-up views of the Id subdomain of the EMCV IRES interacting with uS13, presented in different orientations, with map density rendered at 70% transparency to show the fitting of the atomic model. (G–I) Interaction between protein uS19 (green) of the 40S subunit and the EMCV IRES (red) shown by atomic models presented in different orientations, with interacting residues labeled. (J–L) Close-up views of the Ib and Id subdomains of the EMCV IRES interacting with uS19, presented in different orientations, with map density rendered at 70% transparency to show the fitting of the atomic model.

structure (Fig. 3J–L). GNRA tetraloops constitute the largest class of tetraloops and commonly function as determinants of RNA tertiary structure. GNRA tetraloops bind to three types of RNA receptor: the 11nt receptor (which is specific for GAAA loops, which stack on an adenosine platform in the receptor), the IC3 receptor, another asymmetric internal loop present in the IC3 subgroup of self-splicing introns, and the minor groove receptors, which are formed by two helical base-pairs and interact via A-minor type hydrogen bonding with the third and fourth nucleotides of the tetraloop (Fiore and Nesbitt, 2013; Wu et al, 2012). Several variants of the latter type of tetraloop-receptor have been identified, which deviate with respect to the location of the hydrogen bonding interface, in some instances due to rotation of the tetraloop bases away from the receptor. The direct contact between the EMCV mRNA tetraloop and the TψC arm of initiator tRNA observed here represents a structurally distinct and novel pattern of interactions, which contribute directly to complex stabilization and ribosomal positioning of the IRES—features not previously reported in the context of translation initiation.

Mutational analysis of domain I (Fig. 4A) confirmed that the structure of its apical region is critical for the IRES function. Thus, disruption of Ih2 and Ih4 abrogated 48S complex formation on the IRES in the in vitro reconstituted system, but activity was restored by compensatory mutations (Fig. 4B, lanes 5–8). Disruption of the Ih1 apical base-pair did not abrogate but strongly reduced the activity of the IRES, whereas the compensatory mutation again restored IRES function (Fig. 4B, lanes 3–4). Mutations in the Id loop affected IRES activity to different extents, with deletion of two A residues having the biggest effect (Fig. 4B, lanes 9–12). In accordance with previous reports regarding the importance of the tetraloop for type 2 IRES function (Roberts and Belsham, 1997; López de Quinto and Martínez-Salas, 1997; Robertson et al, 1999; Nateri et al, 2000; Fernández-Miragall and Martínez-Salas, 2003, mutations in the GNRA loop also impaired formation of 48S complexes on the EMCV IRES (Fig. 4C). Consistent with the intimate interaction of the EMCV IRES with the initiator tRNA, 48S complex formation was also sensitive to post-transcriptional modification of initiator tRNA: in vitro transcribed tRNA was less efficient and also promoted a higher level of initiation at the aberrant upstream $AUG_{826}$ independently of the presence/absence of eIF1 and eIF1A (Fig. EV5A).

Interestingly, the interaction of the IRES and uS19 mimics canonical contacts between 28S rRNA of the 60S subunit and uS19 in the classical-1 PRE state (Budkevich et al, 2011; Bhaskar et al, 2020) (Fig. 5A, B), suggesting that the IRES evolved to exploit a pre-existing rRNA/rprotein contact. It also implies that the IRES would sterically clash with the 60S subunit (Fig. 5C) preventing formation of 80S ribosomes (Fig. 5D), and that displacing of the IRES is a prerequisite for ribosomal subunit joining. During the final step of preparation of 48S complexes for ribosomal subunit

joining, eIF5B, which replaces eIF2 after eIF5-induced GTP hydrolysis, promotes repositioning of initiator tRNA to match its orientation in 80S ribosomes which allows placement of the acceptor stem into the P site of the 60S subunit (Brown et al, 2022; Lapointe et al, 2022). Comparison of the position of Met-tRNA$_i^{Met}$ in eIF2-containing 48S complexes assembled on the EMCV IRES (this study) with the position of Met-tRNA$_i^{Met}$ in initiation complexes containing eIF5B (Brown et al, 2022; Lapointe et al, 2022) indicates a 16.7 Å tRNA shift in the latter (Fig. 5E–J). Such an eIF5B-dependent shift would result in a steric clash between the tRNA T-loop and C548 of the EMCV GNRA tetraloop (Fig. 5K), disrupting the stable tetraloop-tRNA contact and providing a mechanism for IRES displacement required for subsequent subunit joining. Consistently, we found that unlike in the case of HCV-like IRESs (Pestova et al, 2008; Terenin et al, 2008), eIF5B could not substitute for eIF2 in 48S complex formation on the EMCV IRES, yielding only trace amounts of initiation complexes (Fig. EV5B, lanes 3 and 8). eIF2D also could not replace eIF2 (Fig. EV5B, lanes 4 and 9) suggesting that like eIF5B, it could not ensure the position of tRNA compatible with its interaction with the GNRA loop of the IRES.

## Mutations in regions outside of the apex of domain I and the JK domain have relatively low influence on the function of the EMCV IRES in 48S complex formation

The importance of the structure of domain I's long central stem has remained obscure. We investigated it by introducing disruptive/stabilizing and shortening/lengthening mutations in its predicted structural elements (Fig. 6A) and assayed the activity of such mutants in 48S complex formation in the in vitro reconstituted system. Shortening or lengthening of the apical helix Ih1 (mutants 1 and 2) noticeably reduced 48S complex formation (Fig. 6B, lanes 3–4), whereas shortening of the lower helix (mutant 3) had a lesser effect (Fig. 6B, lane 5). The activity of the IRES in 48S complex formation was very tolerant to significant changes in the lower part of the central stem, which included disruption, stabilization, truncation or extension of various regions (mutants 4–9) (Fig. 6C, lanes 3–8).

Stabilization of domain L located downstream of JK domain (mutant 10 in Fig. 6A) or its entire deletion or replacement by 2 or 6 nucleotides (Fig. 7A, mutants mL1-4) did not affect the assembly of 48S complexes on the IRES (Fig. 6C, lane 9; Fig. 7B, lanes 3–6).

The importance of the region upstream of domain I for IRES function is contentious (Jang and Wimmer, 1990; Duke et al, 1992), and the structure of this region is not firmly established. Modeling of the type 2 IRESs of Theiler's murine encephalitis virus and Saffold-like cardiovirus suggest that this region forms a pseudoknot (Pilipenko et al, 2001; Drexler et al, 2010), and that the EMCV IRES has the potential to form an analogous structure. We assayed the importance of the presence and structure of this upstream region for function using a panel of mutants (Fig. 7A). Structural disruption (Fig. 7A,

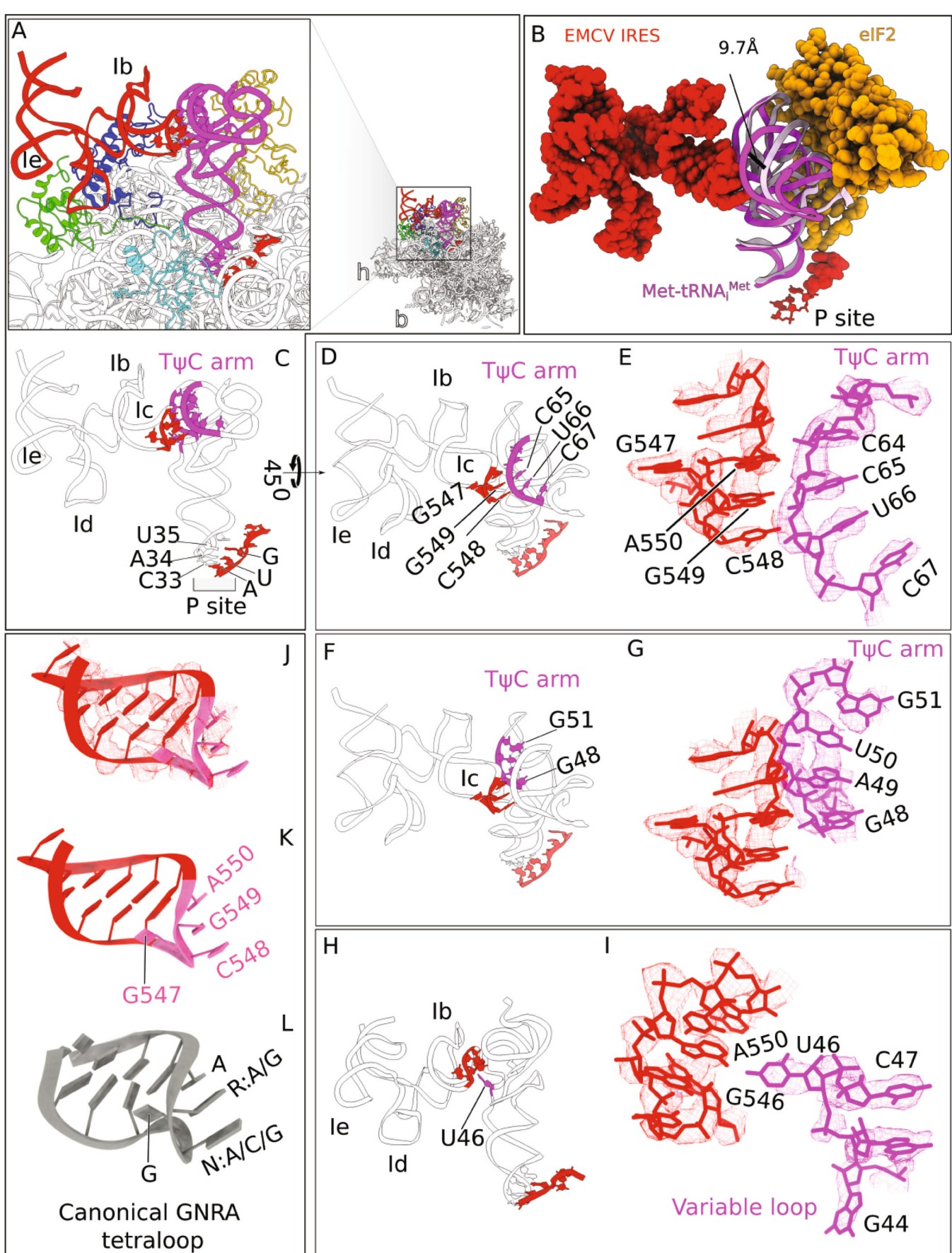

◄  **Figure 3.  The conformation of the EMCV IRES domain I tetraloop and its interactions with initiator tRNA.**

(A) Close-up view of the atomic model of the 40S subunit head region, showing interactions between the EMCV IRES (red), initiator tRNA (magenta), and 40S subunit proteins uS19 (green) and uS13 (blue). (B) Structural comparison reveals that the acceptor arm of the P-site-bound tRNA undergoes a ~9.7 Å displacement from its position associated with eIF2 in canonical 48S complexes (PDB id: 8OZO) (tRNA is in light pink) to a new position where it interacts directly with the EMCV IRES (tRNA is in magenta). (C) Interaction between the tetraloop of Ic domain of the EMCV IRES and initiator tRNA, and codon-anticodon recognition between initiator tRNA at the P site and the EMCV initiation codon AUG$_{834}$. (D–I) structural details of the EMCV IRES tetraloop regions interacting with initiator tRNA. (D, E) Interaction is shown by atomic models and by atomic models fitted into Coulomb density (rendered at 70% transparency, and highlighting the front TψC arm), respectively. (F, G) Same interaction is shown but with a focus on the rear TψC arm. (H, I) Interaction between the variable loop of initiator tRNA and the tetraloop of the EMCV IRES visualized by atomic models and atomic models fitted into Coulomb density (rendered at 70% transparency). (J) Conformation of the EMCV IRES tetraloop, shown by the atomic model fitted into Coulomb density (rendered at 70% transparency). (K, L) Comparison of the tetraloop structure found in the EMCV IRES with canonical GNRA tetraloops (where N can be A/G/C and R is A/G) (PDB: 1ZIF).

mutants 5'm1-2) or deletion of this region (Fig. 7A, mutants 5'm3-5) did not affect 48S complex formation in the in vitro reconstitution system (Fig. 7B, lanes 8–12). Taken together, these and earlier studies (Pestova et al, 1996b; Kolupaeva et al, 1998; Kolupaeva et al, 2003) identified domains I and J-K as being critical for EMCV IRES function in the in vitro reconstituted system, whereas upstream and downstream domains were non-essential.

# Discussion

Our breakthrough cryo-EM structure of 48S complexes reconstituted on the EMCV IRES uncovered essential direct interactions of the apex of domain I of the IRES with the 40S subunit and initiator tRNA. While this manuscript was being reviewed, a lower resolution cryo-EM study of the 48S complexes formed on the EMCV IRES in rabbit reticulocyte lysate reported similar interactions of domain I with initiator tRNA and ribosomal proteins uS13 and uS19 (Das and Hussain, 2025). Regarding the high conservation of primary and secondary structures of the apex of domain I among Type 2 IRES, and particularly of the elements directly involved in the interaction with 40S subunits and initiator tRNA (Appendix Fig. S5, Appendix Table S3) as well as mutagenesis data supporting the importance of this region for the function of these IRESs (López de Quinto and Martínez-Salas, 1997; Roberts and Belsham, 1997; Robertson et al, 1999; Nateri et al, 2000; Fernández-Miragall and Martínez-Salas, 2003; Fernández et al, 2011, 2013), the interactions observed for the EMCV IRES are most likely common for all Type 2 IRESs.

Prior to our study, direct ribosomal interaction was considered as the driving force for initiation only for HCV-, HalV- and CrPV-like IRESs, and the only known essential interaction of Type 2 IRESs involved eIF4G. The direct specific interaction of Type 2 EMCV IRES with the 40S subunit revealed in our study indicates that intimate contact with the 40S subunit is a general common characteristic of diverse initiation mechanisms utilized by different viral IRESs. Interestingly, all of these interactions utilize regions of the 40S subunits that are used for binding of different canonical components of the translational apparatus. Thus, ribosomal binding of CrPV- and HalV-like type 6 IRESs involves A and P sites, respectively (Schüler et al, 2006; Abaeva et al, 2020), whereas HCV-like type 4 IRESs usurp the binding site for the structural core of eIF3 (Hashem et al, 2013). In contrast to HCV-, HalV- and CrPV-like IRESs which do not use all canonical initiation components for 48S complex formation and can therefore usurp and employ their binding sites, initiation on Type 2 IRESs requires initiator tRNA and a full set of eIFs. Thus, ribosomal binding of the EMCV IRES includes interaction with uS19, which interacts with H38 (the A-site finger) of the 60S

subunit to form the B1a intersubunit bridge (Budkevich et al, 2011; Bhaskar et al, 2020), a translational component that does not participate in 48S complex formation.

Although the interaction of Type 2 IRESs with individual 40S subunits has been reported, binding was rather weak (equilibrium dissociation constant ($K_d$) of $55 \pm 10$ nM) (Chamond et al, 2014), compared to $1.9 \pm 0.3$ nM for the HCV IRES (Kieft et al, 2001) and of unclear specificity, since protection from SHAPE modification was observed in domain G (upstream of domain H) and in the nonconserved central region of domain I. Our structure shows that the interaction of the apex of domain I with the 40S subunit is supported and stabilized by the specific interaction of its GNRA loop with the initiator tRNA TψC domain. Interactions of the apex of domain I with 40S subunits and initiator tRNA in 48S ICs pose the key question of whether they can occur independently of the JK domain's interaction with eIF4G/eIF4A and thus act as the primary interactions responsible for the ribosomal recruitment of Type 2 IRESs. Although the interaction of the JK domain with eIF4G/4A could likely aid ribosomal recruitment of Type 2 IRESs, the continuous association of the apex of domain I with initiation complexes even after dissociation of the eIF4G/4A-bound JK domain during grid preparation suggests that it is sufficiently strong and might therefore be the primary interaction responsible for the ribosomal recruitment of Type 2 IRESs.

Binding of the apex of domain I to the 40S subunit creates a steric clash with the 60S subunit indicating that the IRES must be released to allow subunit joining. Strikingly, the interaction of the IRES with initiator tRNA not only represents a contact that could facilitate ribosomal recruitment of the IRES but also provides a potential mechanism for its displacement. Thus, upon dissociation of eIF2 following eIF5-induced hydrolysis of eIF2-bound GTP and subsequent replacement of eIF2 by eIF5B, eIF5B-bound initiator tRNA undergoes a 16.7 Å shift compared to its position in eIF2-containing 48S complexes (Brown et al, 2022; Lapointe et al, 2022), which would result in a steric clash between the tRNA T-loop and the C548 residue of the tetraloop-containing subdomain. This clash could disrupt the tetraloop-tRNA interaction, potentially leading to release of the apex of domain I from the 40S subunit. The observed inability of eIF5B or eIF2D to substitute for eIF2 in 48S complex formation on the EMCV IRES might also be explained by the specific unique position of initiator tRNA in eIF2-containing initiation complexes that is compatible with the interaction with the GNRA loop of the EMCV IRES.

Thus, Type 2 IRESs contain two major elements: the apical region of domain I which binds the 40S subunit and tRNA, and the J-K domain which binds eIF4G/eIF4A. Whereas the primary role of the former

would be consistent with facilitating attachment of the IRES to 43S preinitiation complexes, the primary role of the latter is likely to promote loading of the region around the initiation codon into the mRNA-binding channel, which is supported by the observation that eIF4A/4G induce ATP-dependent conformational changes in this region of Type 2 IRESs (Kolupaeva et al, 2003). The fact that the activity of the EMCV IRES was very tolerant to significant changes in the lower part of the central stem of domain I (disruption, stabilization, truncation or extension of various regions) (Fig. 6) suggests that it might primarily serve as a connector between the two major functional elements, albeit of certain length, configuration and flexibility. Consistent with this idea, the lower part of the central stem of domain I is the least conserved region in Type 2 IRESs (Hellen and Wimmer, 1995; Jackson and Kaminski, 1995). However, some influence of this region on the release of the IRES during ribosomal subunit joining or on potential cooperativity of ribosomal recruitment of domains I and JK cannot be strictly excluded. The region upstream of domain I as well as domain L located downstream of the J-K domain were also not essential for 48S complex formation on the EMCV IRES in the in vitro reconstituted system (Fig. 7), emphasizing further the exceptional role of two major elements: the apical region of domain I and the J-K domain.

The key remaining question in the initiation on Type 2 IRESs is the exact mechanism of action of eIF4G/4A and the role in it of their specific association with the JK domain. To avoid potential artifacts, we did not perform chemical cross-linking of the assembled 48S complexes used for cryo-EM studies, even though cross-linking stabilizes the association eIF3, eIF4A and eIF4G with ribosomal complexes (Brito Querido et al, 2020; Petrychenko et al, 2025). As expected, the structure of uncross-linked 48S complexes did not reveal the position of the eIF4G/4A-bound JK domain. Initiation on the EMCV IRES does not require the portion of eIF4G that is responsible for its interaction with eIF3 (Lomakin et al, 2000; Sweeney et al, 2014), which poses the following questions: (i) does the JK domain play the role in positioning of the eIF4G/eIF4A/JK domain complex by specific interactions with e.g., the 40S subunit or eIF3? and (ii) is the position of eIF4G/eIF4A in initiation complexes assembled on the EMCV IRES the same as in canonical initiation complexes or does the JK domain usurp eIF4G/eIF4A and displace them from their conventional binding site analogously to how the HCV IRES binds eIF3 and displaces it from its position on the 40S subunit (Hashem et al, 2013)? Thus, to elucidate the molecular mechanism of action of eIF4G/4A during initiation on Type 2 IRESs, future studies should focus on obtaining structures of 48S complexes showing the eIF4G/eIF4A-bound JK domain.

## Methods

### Reagents and tools table

| Reagent/Resource | Reference or Source | Identifier or Catalog Number |
|---|---|---|
| **Experimental models** | | |
| Rabbit reticulocyte lysate | Green Hectares | |
| *E. coli* BL21(DE3) | Thermo Fisher Scientific | C600003 |
| *E. coli* DH5α | Thermo Fisher Scientific | 18258012 |
| Native total calf liver tRNA | Promega | Y209X |

| Reagent/Resource | Reference or Source | Identifier or Catalog Number |
|---|---|---|
| **Recombinant DNA** | | |
| pTRM1 (pBR322–tRNA$_i$Met) | Pestova and Hellen (2001) | |
| T7-EMCV(373-1656) wt_pUC57 | This study | |
| T7-EMCV(373-987)wt_pUC57 | This study | |
| pTZ18R-EMCV(315-1160) | This study | |
| pQE(His6-eIF1) | Pestova et al (1998a) | |
| pET (His6-eIF1A) | Pestova et al (1998a) | |
| pET(His6-eIF4A) | Pestova et al (1996a) | |
| pET(His6-eIF4B) | Pestova et al (1996a) | |
| pET28(His6-eIF4G457-1396) | Pestova et al (1996b) | |
| pET28b-nPTB | Pilipenko et al (2001) | |
| pET28a-[*E. coli* Met-tRNA synthetase] | Lomakin et al (2006) | |
| pReceiver-B01-eIF2D | Skabkin et al (2010) GeneCopoeia | EX-W0697-B0 |
| pET (His6-eIF1A-Cys) variants | Yu et al (2009) | |
| **Antibodies** | | |
| **Oligonucleotides and other sequence-based reagents** | | |
| Primers | Life Technologies Corp. | Appendix Table 4 |
| **Chemicals, Enzymes and other reagents** | | |
| T7 RNA polymerase | Thermo Fisher Scientific | EP0111 |
| Sequenase v2.0 DNA polymerase | Thermo Fisher Scientific | 70775Y200UN |
| Avian Myeloblastosis Virus Reverse Transcriptase | Promega | M5108 |
| T4 Polynucleotide Kinase | New England Biolabs | M0201S |
| EcoRI | New England Biolabs | R0101S |
| SpeI | New England Biolabs | R3133S |
| BstNI | New England Biolabs | R0168S |
| RNase T1 | Thermo Fisher Scientific | EN0541 |
| Isopropyl-beta-D-thiogalactoside | Gold Biotechnology | I2481C100 |
| Ni-NTA Agarose | QIAGEN | 30230 |
| MonoS 5/50 GL column | Cytiva Life Sciences | 17516801 |
| MonoQ 5/50 GL column | Cytiva Life Sciences | 17516601 |
| Superdex 200 Increase 5/150 GL | Cytiva Life Sciences | 28990945 |
| NuPAGE 4–12% Bis-Tris gels | Thermo Fisher Scientific | NP0322BOX |
| SimplyBlue Safestain | Thermo Fisher Scientific | LC6065 |
| RiboLock RNase inhibitor | Thermo Fisher Scientific | EO0381 |
| Pierce protease inhibitor tablets | Thermo Fisher Scientific | 32955 |
| Imidazole | Thermo Fisher Scientific | 396745000 |
| 1-(*p*-Bromoacetamidobenzyl) ethylenediamine-*N,N,N′,N′*-tetraacetic acid, iron(III) | Dojindo | F279-10 |

| Reagent/Resource | Reference or Source | Identifier or Catalog Number |
|---|---|---|
| Spermidine Trihydrochloride | Fisher Scientific | AC21510-0050 |
| Copper Quantifoil grids | Quantifoil Micro Tools | |
| SurePAGE 4–12% Bis-Tris gels | Genscript | M00653 |
| MES SDS Running buffer | Genscript | M00677 |
| $[\gamma\text{-}32P]$-ATP | Revitty | BLU002Z |
| $[\alpha\text{-}32P]$-dATP] | Revitty | BLU012Z |
| Amicon 10k filter units | Millipore Sigma | UFC8010 |
| NTP set | Thermo Fisher Scientific | R0481 |
| dNTP set | Thermo Fisher Scientific | 10297018 |
| **Software** | | |
| MotionCor2 | https://emcore.ucsf.edu/ucsf-software | |
| CTFFIND4 | https://grigorieflab.umassmed.edu/ctffind4 | |
| Topaz | https://cb.csail.mit.edu/topaz/ | |
| Relion4.0 | https://relion.readthedocs.io/en/release-4.0/ | |
| CryoSparc | https://cryosparc.com/ | |
| Coot | https://www2.mrc-lmb.cam.ac.uk/personal/pemsley/coot/ | |
| Chimera | https://www.cgl.ucsf.edu/chimera/ | |
| Phenix | https://phenix-online.org | |
| 3dRNA | http://biophy.hust.edu.cn/new/3dRNA | |
| Adobe Illustrator | https://www.adobe.com/products/illustrator.html | |
| Adobe Photoshop | https://www.adobe.com/products/photoshop.html | |
| ImageQuant TL | Cytiva Life Sciences | |
| Clustal Omega | https://www.ebi.ac.uk/Tools/msa/clustalo/ | |
| R-Scape | http://eddylab.org/R-scape/ | |
| **Other** | | |
| Vitrobot Mark IV | Thermo Fisher Scientific | |
| K3 Summit direct electron detector | Gatan, Inc | |
| Tecnai F20 | Thermo Fisher Scientific | |
| Tecnai Polara F30 | Thermo Fisher Scientific | |
| Krios | Thermo Fisher Scientific | |
| Amersham Typhoon IP | Cytiva Life Sciences | 29187194 |
| Akta Pure | Cytiva Life Sciences | |
| Owl Aluminum-Backed Sequencer | Thermo Fisher Scientific | S4S |

## Construction of plasmids

The transcription vector for human tRNA$_i$^Met has been described (Pestova and Hellen, 2001). The EMCV transcription vectors T7-EMCV(373-1656)wt_pUC57 and T7-EMCV(373-987)wt_pUC57, containing a T7 promoter followed by EMCV nt. 373–1656 or EMCV nt. 373–987 (GenBank: M81861.1) inserted into pUC57, and mutants derived from these plasmids were made by GenScript Corp. (Piscataway, NJ) and by Synbio Technologies (Monmouth Junction, NJ), respectively. The EMCV plasmids were linearized by EcoRI, and mRNAs were transcribed using T7 RNA polymerase (Thermo Fisher Scientific). The plasmid pTZ18R-EMCV(315-1160) containing a T7 promoter followed by EMCV nt. 315–1160, with ATG826-828ATT and TGC861-863AGT substitutions in the EMCV sequence to eliminate the weakly used AUG826 (and thus to improve the homogeneity of the 48S complexes) and to add an Spe1 site (for restriction, to yield mRNA that terminates at nt. 862) was used for generating mRNA employed in assembly of 48S complexes for cryo-EM studies, directed hydroxyl radical cleavage experiments and enzymatic footprinting.

Vectors for expression of His$_6$-tagged wild type eIF1 and eIF1A (Pestova et al, 1998a), eIF4A and eIF4B (Pestova et al, 1996a), eIF4G1(653–1599) (Pestova et al, 1996b), nPTB (Pilipenko et al, 2001), *Escherichia coli* methionyl tRNA synthetase (Lomakin et al, 2006), eIF2D (Skabkin et al, 2010), and of eIF1A proteins containing unique surface-exposed cysteines (Yu et al, 2009) have been described.

## Purification of ribosomal subunits, initiation factors and aminoacyl-tRNA synthetases

Native mammalian 40S subunits, eIF2, eIF3, eIF4F, and eIF5B were purified from rabbit reticulocyte lysate (RRL) (Green Hectares, Oregon, WI) as described (Pisarev et al, 2007). Human recombinant eIF1, eIF1A, eIF4A, eIF4B, eIF4G$_{653-1599}$, nPTB, eIF2D and *E. coli* methionyl-tRNA synthetase were expressed in *E. coli* BL21(DE3) (Invitrogen) and purified as described (Skabkin et al, 2010; Pestova et al, 1996a, b, 1998a; Lomakin et al, 2000, 2006).

Native total calf liver tRNA (Promega) and in vitro transcribed tRNA$_i$^Met were aminoacylated using recombinant *E. coli* methionyl-tRNA synthetase (Lomakin et al, 2006; Pisarev et al, 2007).

## Assembly of 48S complexes on the EMCV IRES for cryo-EM studies

48S complexes were assembled by incubating 9 pmol EMCV (ATG826-828ATT/TGC861-863AGT) mRNA with 3 pmol 40S subunits, 12 pmol eIF2, 6 pmol eIF3, 6 pmol eIF4F, 12 pmol eIF4A, 10 pmol eIF4B, 15 pmol eIF1, 15 pmol eIF1A, 10 pmol nPTB and 5 pmol native Met-tRNA$_i$^Met in A (20 mM Tris pH 7.5, 100 mM KCl, 2.5 mM MgCl$_2$, 2 mM DTT and 0.25 mM spermidine) supplemented with 1 mM ATP and 0.4 mM GTP in a final volume of 40 μl for 15 min at 37 °C.

## Preparation of cryo-EM grid and image acquisition

Copper Quantifoil grids R0.6/1.0 with mesh size 300 (Quantifoil Micro Tools GmbH) were subjected to glow discharge with air for 30 s using a PELCO easiGlow cleaning system set to a plasma

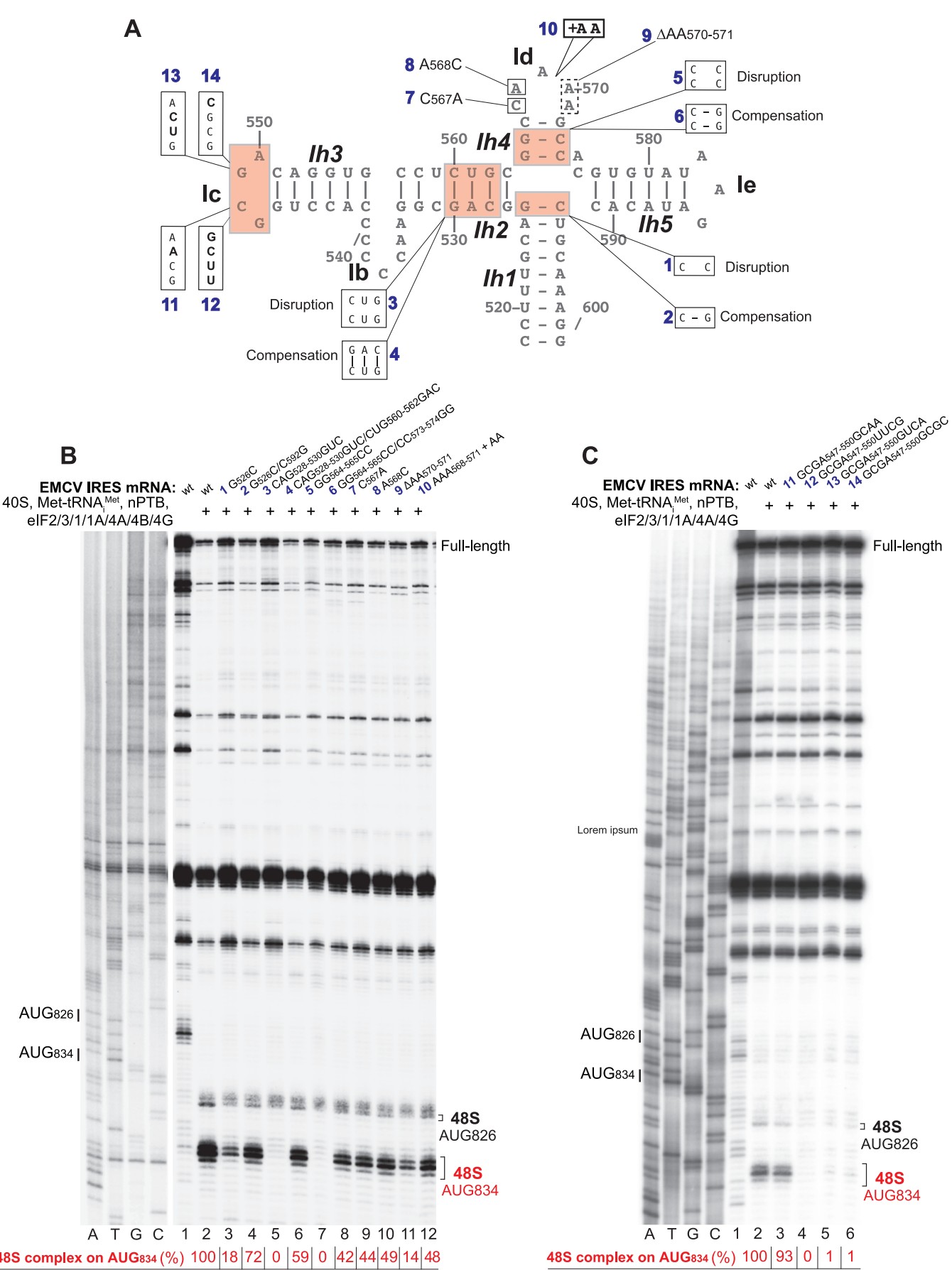

◀

**Figure 4. Influence of mutations in the apex of domain I on 48S complex formation on the EMCV IRES.**

(A) Secondary structure model of the apex of domain I showing introduced mutations. (B, C) Toe-printing analysis of 48S complex formation on wt and mutant EMCV IRES mRNAs in the presence of individual 40S subunits, Met-tRNA$_i^{Met}$, nPTB and indicated eIFs. The efficiency of 48S complex formation on AUG$_{834}$ (red numbers) was quantified taking the efficiency of complex formation on the *wt* EMCV IRES mRNA as 100%. Standard deviations (omitted for clarity) did not exceed 6%. Lanes C, T, A, and G depict the wt EMCV sequence generated using the same primer. The positions of initiation codons are indicated on the left and of assembled 48S complexes on the right. The division between sequence lanes and lanes 1–12 in (A) indicates that these two sets of lanes were derived from the same gel, exposed for different lengths of time. Source data are available online for this figure.

current of 15 mA, to make the carbon film surface negatively charged, which allows buffer solutions to spread easily. These grids were immediately placed into the Vitrobot Mark IV (Thermo Fisher Scientific, Carlsbad, CA) maintained at 4 °C and 98% humidity. 3 μl of the above-mentioned sample solution was applied onto the grid and blotted for 4 s with blot force 3. The sample containing grid was plunged into liquid ethane and then carefully transferred to liquid nitrogen. Grids were clipped and screened to confirm sample distribution and ice thickness using a Tecnai F20 electron microscope (Thermo Fisher Scientific) equipped with a field emission gun (FEG) operating at 200 kV and a K3 Summit direct electron detector (Gatan, Inc, Pleasanton, CA).

After screening the best grids are selected and further data was collected on a Tecnai Polara F30 and Titan Krios (Thermo Fisher Scientific) microscopes operated at 300 KeV equipped with a K3 direct detector (Gatan). Movies were collected at a pixel size of 0.95 Å/pixel for Tecnai Polara F30 and 0.83 Å/pixel for Krios. The defocus range was set from −1 to −2.5 μm. A total of 7733 micrographs were collected. 40S ribosomal subunits showed a preferred orientation, so, portions of the data were collected with a 35° stage tilt.

## Image processing

A workflow of the data processing for the cryo-EM image was shown in Fig. EV3. Drift, gain correction, and dose weighting were performed using MotionCor2 (Zheng et al, 2017) with a local patch correction with 7 × 5 patches. The contrast transfer function (CTF) of each micrograph was estimated using CTFFIND4 (Rohou and Grigorieff, 2015). Particle picking was performed using Topaz (Bepler et al, 2019). Good particles were selected by 2D classification and trained using 20,000 particles in Relion4.0 (Scheres, 2012, 2016; Zivanov et al, 2018, 2019). Autopicking using the trained Topaz model yielded 835,875 particles. Particles picked by Topaz were subjected to four rounds of 2D classification for a further selection of good particles, which yielded 688,588 particles. All particles were pooled together and used for initial model generation followed by 3D auto-refinement, applying C1 symmetry in Relion4.0. CTF refinements were done to correct for magnification anisotropy, fourth-order aberrations, per-particle defocus, and per-particle astigmatism, followed by another 3D auto-refinement. The final resolution after refinement was 3.2 Å (Appendix Fig. S1). Then 3D classification was performed without alignment, using the angular information from the previous refinement step. Four distinct classes—Class I (134,626 particles), Class II (98,748 particles), Class III (89,242 particles), and Class IV (72,526 particles)—were identified, all exhibiting clear density for the 40S ribosomal subunit, eIF1A and the IRES bound to the inter-subunit face of the 40S head. Classes II–IV also contained density for eIF1,

whereas not fully resolved Class I contained additional density for eIF2, Met-tRNA$_i^{Met}$, and blurry density for eIF3. Additionally, Classes II–IV differ from one another in the extent of 40S head rotation. Focused classification of the Class I population yielded a refined subclass, termed Class Ia, comprising 59,153 particles with well-resolved density for the 40S, eIF2, eIF3, eIF1A, Met-tRNA$_i^{Met}$, and the IRES. We further did multiple rounds of focused refinement on the 40S head along with associated EMCV mRNA and Met-tRNA$_i^{Met}$ regions, and this yielded a 3.1 Å reconstructed map. In addition, series of focused 3D classification, 3D Flexible Refinement (https://guide.cryosparc.com/processing-data/tutorials-and-case-studies/tutorial-3d-flex-mesh-preparation), Blush regularization (Kimanius et al, 2024) on EMCV mRNA, 40S-head, and Met-tRNA$_i^{Met}$ regions and followed by relion_reconstruct with Ewald sphere correction were performed to resolve the tetraloop-tRNA-40S-head interacting region. All raw FSC curves representing the resolution estimations are shown in Appendix Fig. S1.

## Model building and refinement

For starting models we used available models for the 48S IC (PDB: 7SYR) (Brown et al, 2022) and eIF3 (PDB: 5A5T) (des Georges et al, 2015). Initial rigid body model fitting was performed using UCSF Chimera v1.1 (Pettersen et al, 2004) with additional manual flexible fitting in Coot (Emsley and Cowtan, 2004). All models were further flexibly fitted using Phenix geometry minimization and multiple rounds of PHENIX real-space refinement (Adams et al, 2010; Afonine et al, 2018). The initial model for the EMCV domain-I (residues 518–600) was built by using 3dRNA (Zhang et al, 2022) restraints based on the connectivity of the bases from the 2D structure).

## Determining EMCV mRNA sequence nucleotide identity

The EMCV genome (GenBank: M81861.1; https://www.ncbi.nlm.nih.gov/nuccore/M81861.1) contains an IRES sequence of 441 nucleotides in length covering bases 396UUC…AUG836. The density of IRES visible in our map is approximately 80–90 nucleotides and so we had to determine which region of the complete IRES was visible. To determine which region of the IRES is visible in our density maps (Fig. EV4B), we first identified the highest resolution portion, a loop bound into the pocket between alpha helix 2–3 of ribosomal protein uS13 and the beta-sheet of uS19. Here we could identify sequential purine (R) and pyrimidine (Y) bases (Fig. EV4C). This region of the IRES contained 14 nucleotides where their identity could be determined giving a putative sequence of YRYRRYNNNRNNYY. Since the directionality of the sequence cannot be determined at this resolution, we performed the following search in both the 5′ to 3′ and 3′ to 5′

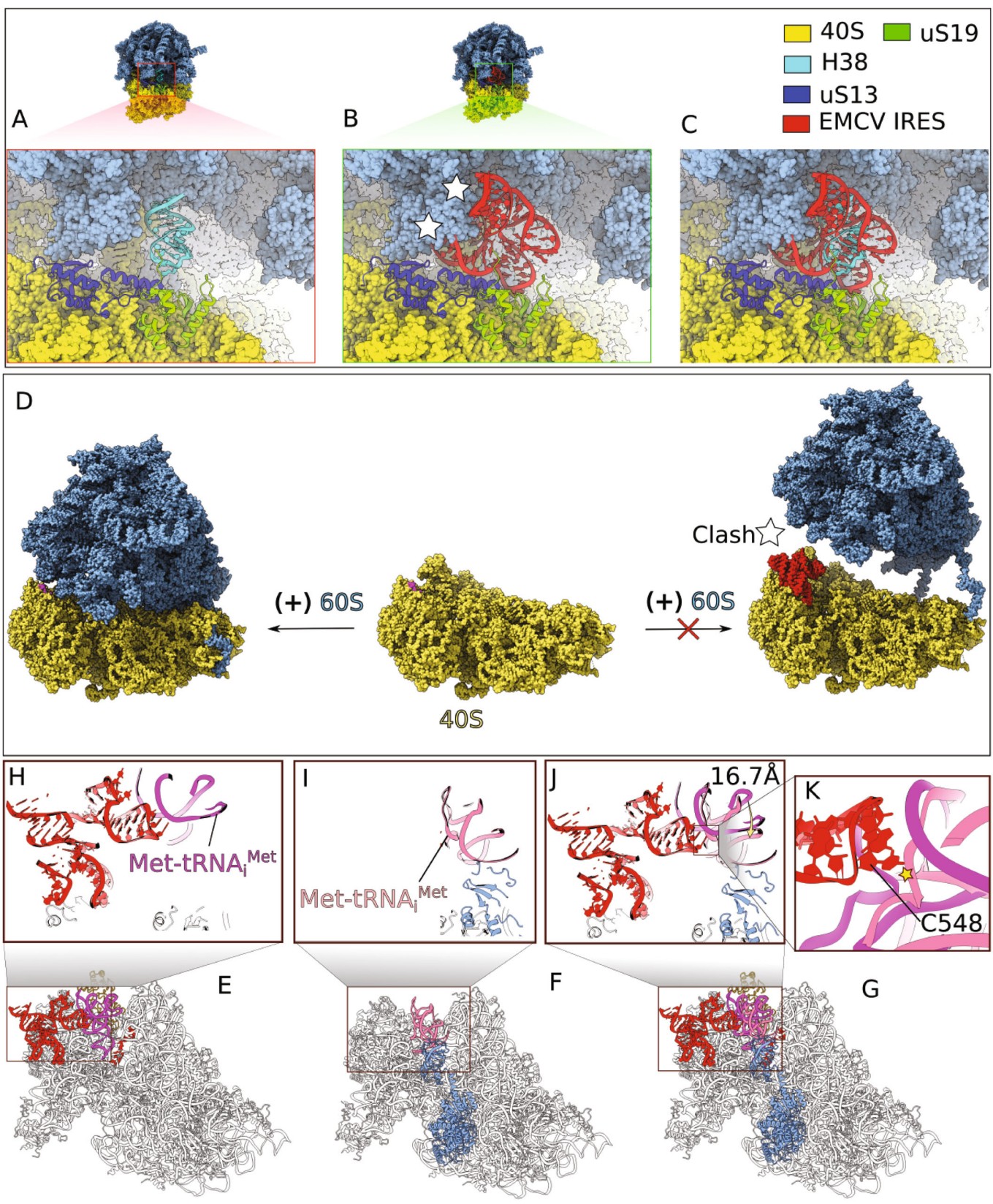

**Figure 5.  Release of the EMCV IRES from the 40S ribosomal head is necessary for joining of the 60S subunit and formation of the 80S ribosome.**

(A) Close-up view of the 80S complex, showing that its formation is stabilized by the inter-subunit bridge B1a, which is established between H38 of the 60S subunit and uS19 of the 40S subunit. (B) Close-up view on a steric clash between the EMCV IRES and the 60S subunit. (C) The EMCV IRES appears to compete for the same binding site on uS19 that H38 occupies during 80S formation. (D) On the right, formation of the 80S complex is prevented by a steric clash between the 40S-bound EMCV IRES (red) and the 60S subunit. Release of the EMCV IRES is required to prevent this clash. (E) Interactions among the 40S subunit, the EMCV IRES (red), initiator tRNA (magenta) and eIF2α (yellow) (this study). (F) Interactions between the 40S subunit, initiator tRNA and eIF5B after dissociation of eIF2 during subunit joining (pdb id: 7SYV). (G) Superimposition of 40S complexes shown in (E) and (F). (H) Zoomed-in view of the ribosomal positioning of initiator tRNA bound to the EMCV IRES and eIF2α in 48S initiation complexes. (I) Close-up of the ribosomal positioning of initiator tRNA and eIF5B in 40S ribosomal complexes in the process of subunit joining. (J) Superimposition of (H) and (I) showing a 16.7 Å shift in the position of tRNA, which leads to (K) steric interference between the tRNA T-loop and the C548 residue of the EMCV IRES tetraloop-containing subdomain after eIF2 is replaced by eIF5B, disrupting the stable tetraloop-tRNA contact.

directions (e.g., (5′)YRYRRYNNNRNNYY(3′) and (3′)YRYR-RYNNNRNNYY(5′)). As will be detailed below, the 5′ to 3′ case proved to be correct with the observed density corresponding to 561TGCGGCCAAAAGCC574, so it will be explained first, but the 3′ to 5′ details will also follow.

## The (5′)RNRR(3′) direction

The leading RYRR region contained the most consecutive purines, which given their size could be most confidently identified, and was used as an initial search term to find candidate matches within the IRES gene. Using the minimal search term of RNRR we identified 40 locations (excluding overlapping regions) that were present across all domains of the IRES. There were four matches in domain H (402ACGA405; 414AGGG417; 435AAAG438; 443GCAA446), 24 in domain I (462GTGA465; 466AGGA469; 471GCAG474; 482GGAA485; 493GAAG496; 497ACAA500; 501ACAA504; 510GTAG513; 523GCAG526; 527GCAG530; 532GGAA535; 547GCGA550; 562GCGG565; 568AAAA571; 581ATAA584; 594GCAA597; 600GCGG603; 605ACAA608; 624GTGA627; 633ATAG636; 639GTGG642; 643AAAG646; 653ATGG656; 676ACAA679), five in domain J (680GGGG683; 686GAAG689; 697AGAA700; 713ATGG717; 728GGGG731), two in domain K (764GAGG767; 770AAAA773), two in domain L (795ACGG798; 803GTGG806), and three in the unstructured region containing the start codon (816GAAA819; 823ACGA826; 829ATAA832). Narrowing down the potential matches was done in two ways, either by correlating with the known secondary structure of the EMCV IRES, or by increasing the search term to include all confidently identified nucleotides. Both methods independently return the same result.

## Using the IRES secondary structure

The visible sequence (YRYLOOPRRYNNNRNNYYLOOP) contains a loop and so using the EMCV IRES secondary structure we can eliminate matches that occur within helices or regions that contain loops that are too short or too long to match the observed density. The loop region contains 11 nucleotides (Fig. EV4C). So we examined the EMCV secondary structure and found 10 candidates that were within a loop region: one in domain H (414AGGG417), eight in domain I (510GTAG513; 532GGAA535; 547GCGA550; 562GCGG565; 568AAAA571; 581ATAA584; 600GCGG603; 605ACAA608), and one in domain L (795ACGG798). All matches in the unstructured region containing the start codon were excluded. Of the 10 candidate locations, one would require a loop that is too long, 17 nucleotides for 600GCGG603, and the other nine loops that are too short, seven nucleotides for 532GGAA535

and 605ACAA608, six nucleotides for 795ACGG798, four nucleotides for 510GTAG513, three nucleotides for 581ATAA584, and two nucleotides for 547GCGA550 and 568AAAA571. None of these were appropriate even after considering ambiguity in the number of observed nucleotides as the longest would require an error of (+ 6) nucleotides (600GCGG603) and the next shortest an error of (−4) nucleotides (532GGAA535; 605ACAA608), something that is not supported by the observed data. The remaining two locations that were in a region of the IRES that could support a loop of 11 nucleotides were 414AGGG417 in domain H and 562GCGG565 in domain Id. Distinguishing between these two locations can be completed using different methods. First, the global arrangement of the IRES suggests a cloverleaf conformation, something that is absent in domain H but present in domain Id (Fig. EV1). Alternatively, considering the second nucleotide and assuming it's accurately identical to a pyrimidine (e.g., RYRR) this also eliminates domain H (414AGGG417) and leaves only domain Id (562GCGG565).

## Using a longer search term

Disregarding the constraints given by the secondary structure we can also identify domain Id as the location of the IRES density by increasing the length of the search term. We used every combination from RNRR to the full set of visible nucleotides (all terms and results given in Appendix Table S2) and found that YRYRRYNNNRNNYY corresponds with 561TGCGGCCAAAAGCC574 in domain Id. No other region in the IRES had a positive result for the complete search term.

## The (3′)RNRR(5′) direction

Given the resolution, it is not possible to determine the directionality of the high-resolution loop region and so we also searched in the 3′ to 5′ direction. Searching for RRNR we identified 44 locations (excluding overlapping regions) that were present across all domains of the IRES. There were six matches in domain H (401GACG404; 405AGCA408; 414AGGG417; 435AAAG438; 440AATG443; 445AAGG448), 23 in domain I (456AATG459; 464GAAG467; 468GAAG471; 482GGAA485; 493GAAG496; 500AACA503; 512AGCG515; 526GGCA529; 532GGAA535; 546GGCG549; 553GGTG556; 568AAAA571; 583AAGA586; 596AAAG599; 602GGCA605; 613AGTG616; 632GATA635; 641GGAA644; 645AGAG648; 652AATG655; 666AGCG669; 675AACA678; 679AGGG682), six in domain J (686GAAG689; 690GATG693; 697AGAA700; 701GGTA704; 716GGGA719; 728GGGG731), four in domain K (736GGTG739; 764GAGG767; 770AAAA773; 774AACG777), one in domain L (797GGGG800),

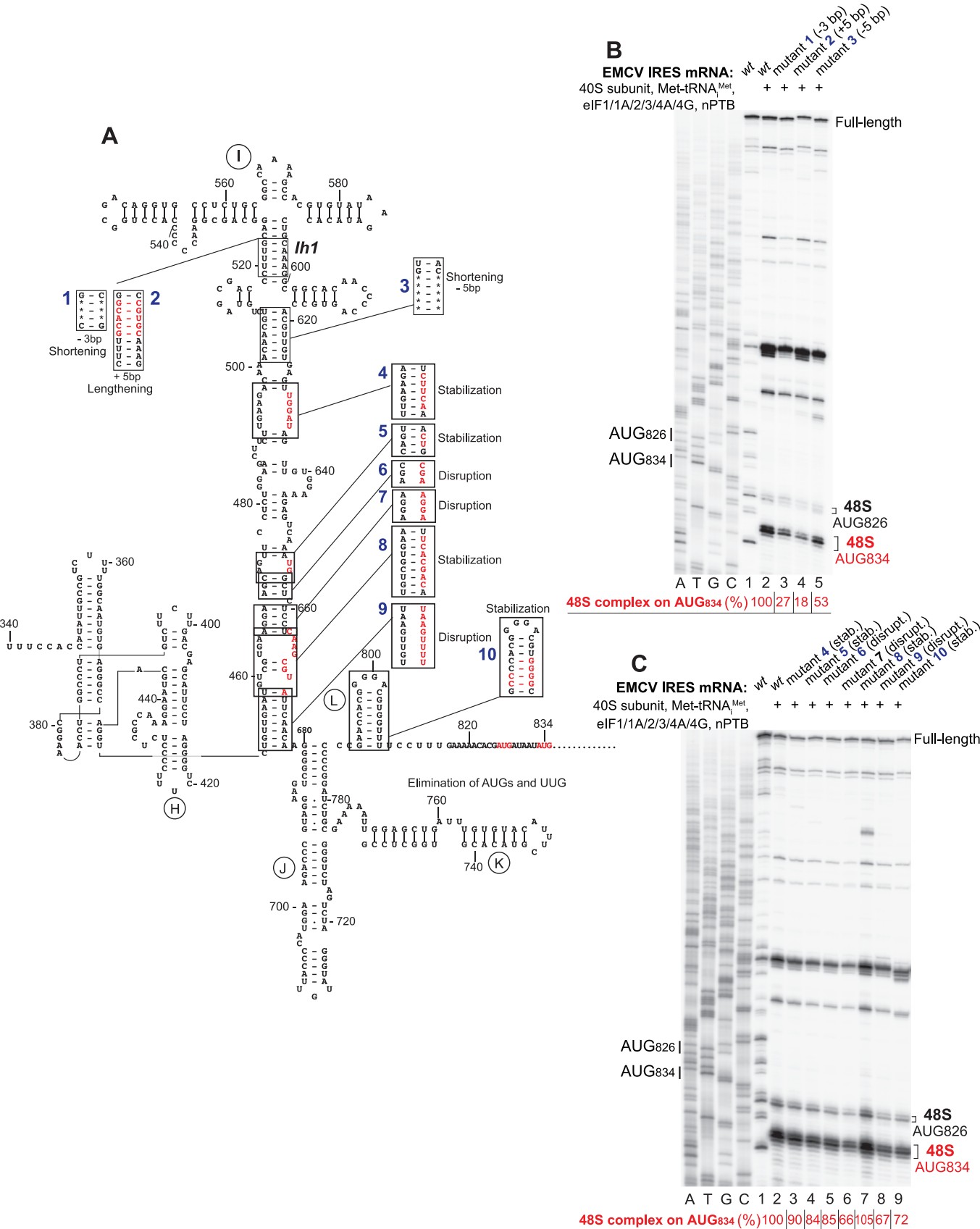

◄

**Figure 6.  Influence of mutations in the stem of domain I on 48S complex formation on the EMCV IRES.**

(A) Secondary structure model of the EMCV IRES showing introduced mutations. (B, C) Toe-printing analysis of 48S complex formation on wt and mutant EMCV IRES mRNAs in the presence of individual 40S subunits, Met-tRNA$_i^{Met}$, nPTB and indicated eIFs. The efficiency of 48S complex formation on AUG$_{834}$ (red numbers) was quantified taking the efficiency of complex formation on the *wt* EMCV IRES mRNA as 100%. Standard deviations (omitted for clarity) did not exceed 8%. Lanes C, T, A, and G depict the wt EMCV sequence generated using the same primer. The positions of initiation codons are indicated on the left and of assembled 48S complexes on the right. Source data are available online for this figure.

and four in the unstructured region containing the start codon (816GAAA819; 820AACA823; 825GATG828; 831AATA834). Narrowing down the potential matches was done in two ways, either by correlating with the known secondary structure of the EMCV IRES, or by increasing the search term to include all confidently identified nucleotides. Both methods independently eliminated all results and therefore strongly suggested that the correct directionality is not (5′)RRNR(3′) but the (5′)RNRR(3′) described in detail above.

## RRNR: using the IRES secondary structure

The visible sequence (LOOPYYNNRNNNYRRLOOPYRY) contains a loop and so using the EMCV IRES secondary structure we can eliminate matches that occur within helices or regions that contain loops that are too short or too long to match the observed density. The loop region contains 11 nucleotides (Fig. EV4C) and so we examined the EMCV secondary structure and found 5 candidates that were within a loop region: three in domain I (512AGCG515; 553GGTG556; 568AAAA571), one in domain J (716GGGA719), and one in domain L (797GGGG800). These would require loops of length 13 (716GGGA719), 12 (553GGTG556), and two (512AGCG515; 568AAAA571; 797GGGG800). The region in domain Ic (553GGTG556) that could be linked with a loop of length 12 (so only an error of one nucleotide) bears closer examination.

## RRNR: using a longer search term

Increasing the length of the search term eliminates all matches (all terms and results given in Appendix Table S2) so that the maximum search YYNNRNNNYRRYRY returns zero matches.

## Model building IRES region 518CC..GG600

Once we had determined the identity of the 561TGCGGCCAAA AGCC574 region we were able to build domains Id, Ic, Id, and Id spanning 83 nucleotides (518CC..GG600) and accounting for 17% of the IRES.

## Analysis of the activity of EMCV IRES mutants in 48S complex formation in the in vitro reconstituted system

48S complexes were assembled by incubating 0.3 pmol of *wt* or mutant EMCV IRES mRNAs with 1 pmol 40S subunits, 1.5 pmol native or in vitro transcribed Met-tRNA$_i^{Met}$ and indicated combinations of 5 pmol eIF2, 2 pmol eIF3, 8 pmol eIF4G$_{653-1599}$, 8 pmol eIF4A, 3 pmol eIF4B, 10 pmol eIF1, 10 pmol eIF1A, 5 pmol nPTB, 8 pmol eIF5B and 8 pmol eIF2D in buffer A (20 mM Tris pH 7.5, 100 mM KCl, 2.5 mM MgCl$_2$, 2 mM DTT and 0.25 mM spermidine) supplemented with 1 mM ATP and 0.4 mM GTP in a

final volume of 20 μl for 15 min at 37 °C. Formation of 48S complexes was monitored by toe-printing using AMV reverse transcriptase (Promega) and $^{32}$P-labeled primer (5′-CGGTATTG-TAGAGCAG-3′) complementary to EMCV nt. 901–916 or (5′-GCAGGTAAAATCCATTACGG-3′) complementary to EMCV nt. 914–933 (Appendix Table S4) (Pisarev et al, 2007). Radiolabeled cDNAs were phenol-extracted, ethanol-precipitated, resolved on 6% polyacrylamide gel and analyzed by Phosphoimager. Experiments were repeated at least 2–3 times, and, where indicated, the intensities of bands corresponding to 48S complexes were quantified using ImageQuant.

## Directed hydroxyl radical cleavage

eIF1A Cys mutant proteins were derivatized with Fe(II)-BABE by incubating 3000 pmol eIF1A with 1-mM Fe(II)-BABE in buffer B (80 mM HEPES, 300 mM KCl, 10% glycerol) in a final volume of 100 μl for 30 min at 37 °C as described (Yu et al, 2009). Derivatized proteins were separated from unincorporated reagent by buffer exchange on Amicon 10k filter units and stored at −80 °C.

48S/[Fe(II)-BABE]-eIF1A complexes were formed by incubating 0.6 pmol EMCV(ATG826-828ATT/TGC861-863AGT) mRNA with 2 pmol 40S subunits and 20 pmol [Fe(II)-BABE]-eIF1A, 3 pmol Met-tRNA$_i^{Met}$, 10 pmol eIF2, 4 pmol eIF3, 20 pmol eIF1, 16 pmol eIF4G$_{653-1599}$, 16 pmol eIF4A, 6 pmol eIF4B and 10 pmol nPTB for 15 min at 37 °C in 40 μl buffer A with reduced DTT (20 mM Tris, pH 7.5, 0.25 mM spermidine, 2.5 mM MgCl$_2$, 0.1 mM DTT) supplemented with 1 mM ATP, 0.5 mM GTP and 30 U RNAse inhibitor for 15 min at 37 °C. To generate hydroxyl radicals, reaction mixtures are incubated on ice for 10 min and then supplemented with 0.06% H$_2$O$_2$ and 5 mM ascorbic acid and incubated on ice for 10 min (Yu et al, 2009). Reactions were quenched by adding 20 mM thiourea. mRNA was then phenol-extracted, ethanol-precipitated and analyzed by primer extension using AMV reverse transcriptase and a γ$^{32}$P-end-labeled primer (5′-GCCCCTTGTTGAATACGCTT-3′) complementary to EMCV nt. 665–684 (Appendix Table S4). Resulting radiolabeled cDNAs were also phenol-extracted, ethanol-precipitated, resolved on 6% polyacrylamide gel and analyzed by Phosphoimager.

## Enzymatic footprinting

48S and RNP complexes were assembled by incubating 0.6 pmol EMCV(ATG826-828ATT/TGC861-863AGT) mRNA with various combinations of 2 pmol 40S subunits, 3 pmol Met-tRNA$_i^{Met}$, 10 pmol eIF2, 4 pmol eIF3, 20 pmol eIF1, 20 pmol eIF1A, 16 pmol eIF4G$_{653-1599}$, 16 pmol eIF4A, 6 pmol eIF4B and 10 pmol nPTB in buffer A supplemented with 1 mM ATP, 0.5 mM GTP and 30 U RNAse inhibitor in a final volume of 40 μl for 15 min at 37 °C. Assembled 48S complexes were either left in the reaction mixture

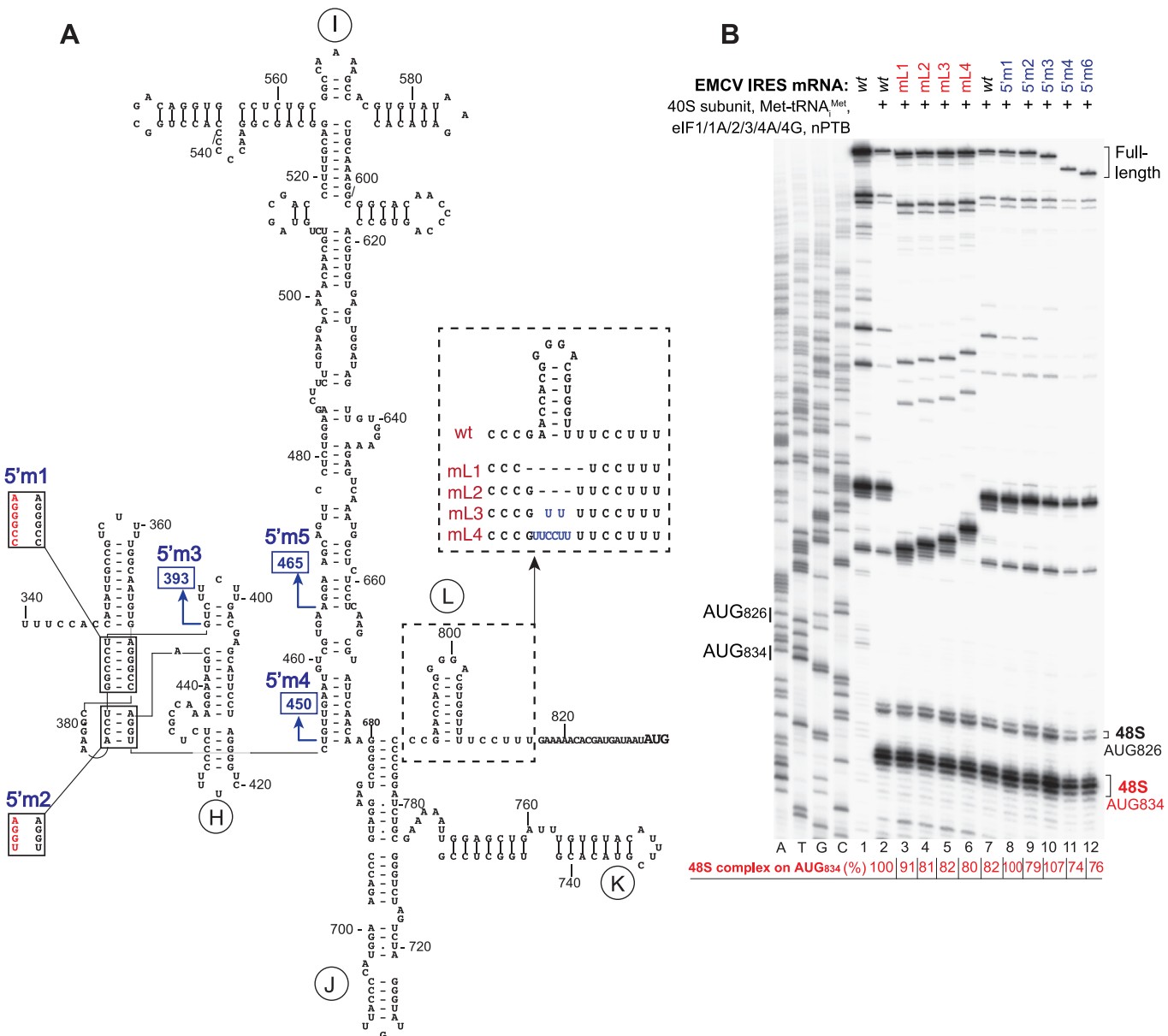

**Figure 7. Influence of mutations in the region upstream of domain I and in domain L downstream of the JK domain on 48S complex formation on the EMCV IRES.**

(A) Secondary structure model of the EMCV IRES showing introduced mutations. (B) Toe-printing analysis of 48S complex formation on *wt* and mutant EMCV IRES mRNAs in the presence of individual 40S subunits, Met-tRNA$_i^{Met}$, nPTB and indicated eIFs. The efficiency of 48S complex formation on AUG$_{834}$ (red numbers) was quantified taking the efficiency of complex formation on the *wt* EMCV IRES mRNA as 100%. Standard deviations (omitted for clarity) did not exceed 6%. Lanes C, T, A, and G depict wt EMCV sequence generated using the same primer. The positions of initiation codons are indicated on the left and of assembled 48S complexes on the right. Source data are available online for this figure.

or purified by sucrose density gradient centrifugation essentially as described (Zinoviev et al, 2015). Thus, 48S complexes are assembled as described above in scaled-up 300 µl reaction mixtures and analyzed by centrifugation through 10–30% sucrose density gradients (SDGs) prepared in buffer A in a Beckman SW55 rotor at 53,000 rpm for 105 min at 4 °C. The optical density of fractionated gradients was measured at 260 nm.

Individual mRNA and assembled 48S and RNP complexes were then enzymatically digested with RNase T1 as described (Kolupaeva et al, 1996). mRNA was phenol-extracted, ethanol-precipitated, and analyzed by primer extension using AMV reverse transcriptase and γ$^{32}$P-end-labeled primers (5'-GCAAGTCTCTTGTTCCATGG-3') complementary to EMCV nt. 844–863 and (5'-GCCCCTTG TTGAATACGCTT-3') complementary to EMCV nt. 665–684 (Appendix Table S4). The resulting radiolabeled cDNAs were also phenol-extracted, ethanol-precipitated, resolved on 6% polyacryla-mide gel and analyzed by Phosphoimager. Experiments were repeated at least 3 times, and, where indicated, the intensities of bands corresponding to T1-induced cleavage were quantified using ImageQuant.

## Determination of a consensus structure for the apical region of domain I

RNA sequences corresponding to the apex of domain I of type 2 IRESs from members of 12 genera of *Picornaviridae* and from several unclassified picornaviruses (Appendix Table S3) were aligned using Clustal Omega. The resulting aligned sequences (41.3–97.5% nucleotide sequence identity) were analyzed using R-Scape (Rivas et al, 2017) to identify statistically significant nucleotide sequence covariation (*E*-value: 0.05) and power. Cacofold (Rivas, 2020) was then used to build a consensus structure.

## Data availability

All cryo-EM maps and molecular models generated in this study have been deposited in the Electron Microscopy Data Bank (EMDB) and in the Protein Data Bank (PDB) with accession codes: EMD-40769, EMD-40770, EMD-40771, EMD-40772, EMD-40773, EMD-40774 and for the corresponding atomic model, PDB 8SUP (EMCV IRES/48S initiation complex). Plasmids generated for this study are available by request to the lead contact. For any inquiries, contact Tatyana Pestova (tatyana.pestova@downstate.edu).

The source data of this paper are collected in the following database record: biostudies:S-SCDT-10_1038-S44318-026-00735-x.

## Peer review information

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

## Acknowledgements

We thank Swastik De and Jeevan GC for providing valuable suggestions. All data were collected at the Cryo-EM core of Columbia University. We thank Robert A. Grassucci, Zhening Zhang, and Yen-Hong Kao for their help with the cryo-EM data collection. This work was supported by National Institutes of Health grants R01GM29169 and R35GM139453 to JF, R35GM122602 to TVP, and R01GM097014 and R21AI188505 to CUTH.

## Author contributions

**Sayan Bhattacharjee**: Data curation; Formal analysis; Validation; Investigation; Visualization; Writing—original draft. **Irina S Abaeva**: Resources; Data curation; Formal analysis; Investigation; Writing—review and editing. **Zuben P Brown**: Data curation; Formal analysis; Validation; Investigation; Visualization; Writing—original draft. **Yani Arhab**: Resources; Data curation; Formal analysis; Investigation; Writing—review and editing. **Hengameh Fallah**: Investigation. **Christopher U T Hellen**: Conceptualization; Resources; Data curation; Supervision; Funding acquisition; Validation; Methodology; Writing—original draft; Project administration. **Joachim Frank**: Conceptualization; Resources; Data curation; Supervision; Funding acquisition; Validation; Methodology; Writing—original draft; Project administration. **Tatyana V Pestova**: Conceptualization; Resources; Data curation; Supervision; Funding acquisition; Validation; Methodology; Writing—original draft; Project administration.

Source data underlying figure panels in this paper may have individual authorship assigned. Where available, figure panel/source data authorship is listed in the following database record: biostudies:S-SCDT-10_1038-S44318-026-00735-x.

## Disclosure and competing interests statement

The authors declare no competing interests.

# Expanded View Figures

**Figure EV1. Secondary structure of the EMCV IRES.**

The EMCV IRES secondary structure model featuring individual domains, the region corresponding to cryo-EM density (red), and specific binding of eIF4A/4G to the JK domain.

▶

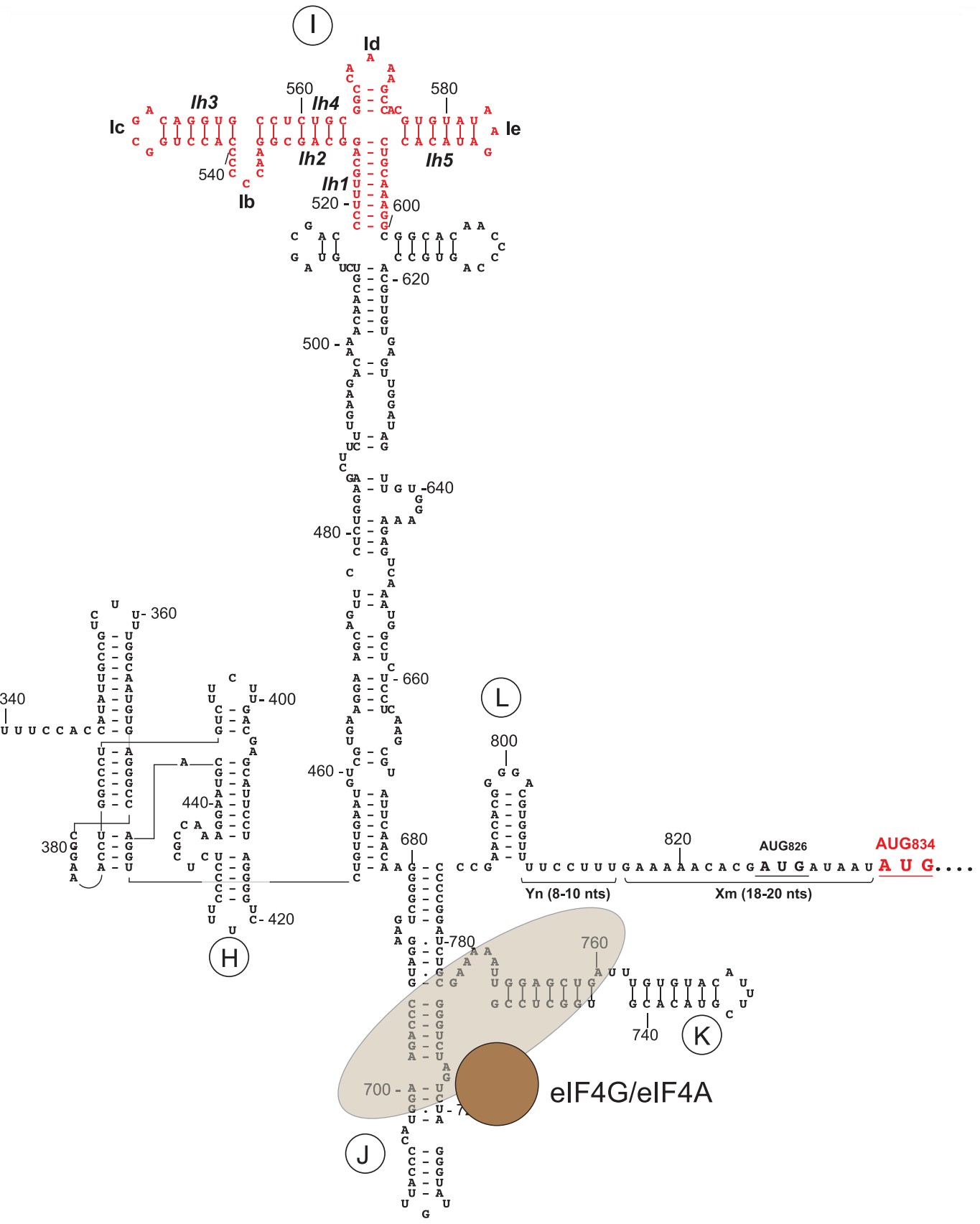

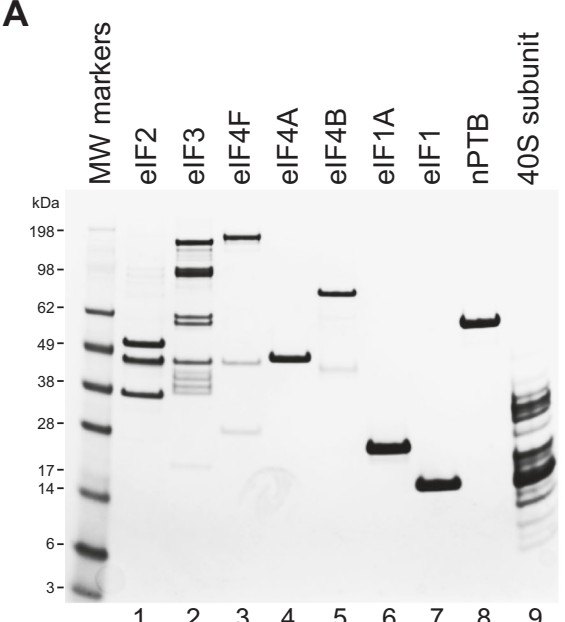

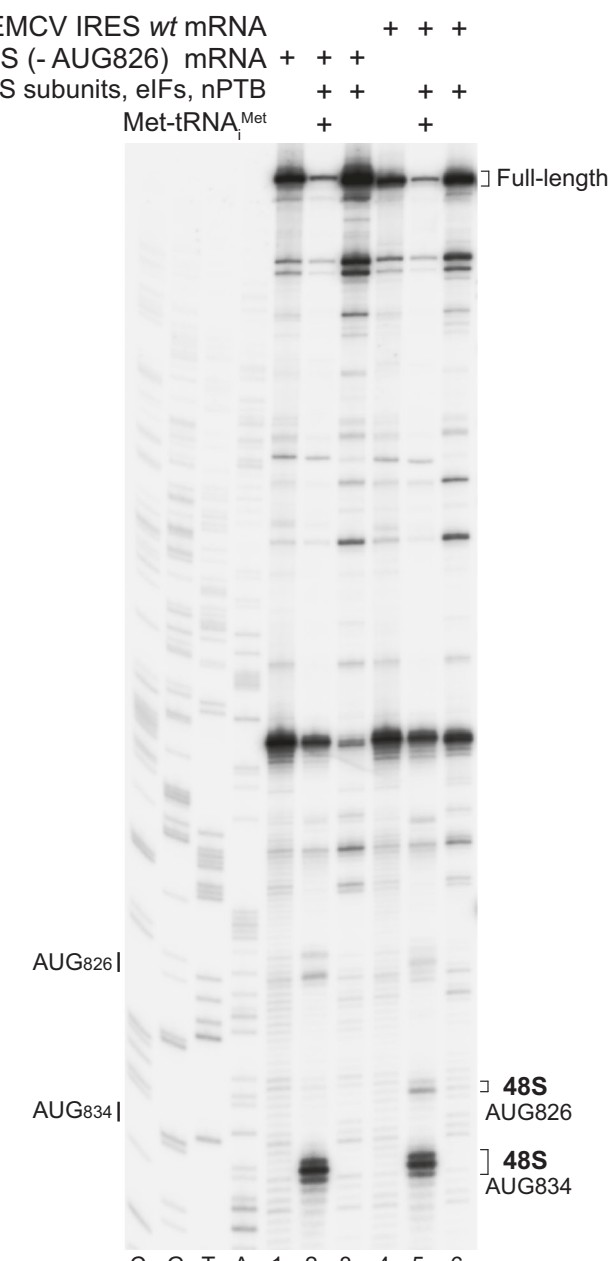

**Figure EV2.  48S complex formation on the *wt* EMCV IRES and the IRES with mutated AUG₈₂₆.**

(**A**) Purified initiation factors, nPTB and 40S subunits, analyzed by SDS-PAGE followed by SimplyBlue staining. (**B**) Toe-printing analysis of 48S complex formation on *wt* EMCV IRES mRNA and IRES mRNA with mutated AUG₈₂₆ in the presence of 40S subunits, initiator Met-tRNAᵢᴹᵉᵗ, nPTB and indicated initiation factors. Lanes C, T, A, and G depict wt EMCV sequence generated using the same primer. The positions of initiation codons are indicated on the left and of assembled 48S complexes on the right. Source data are available online for this figure.

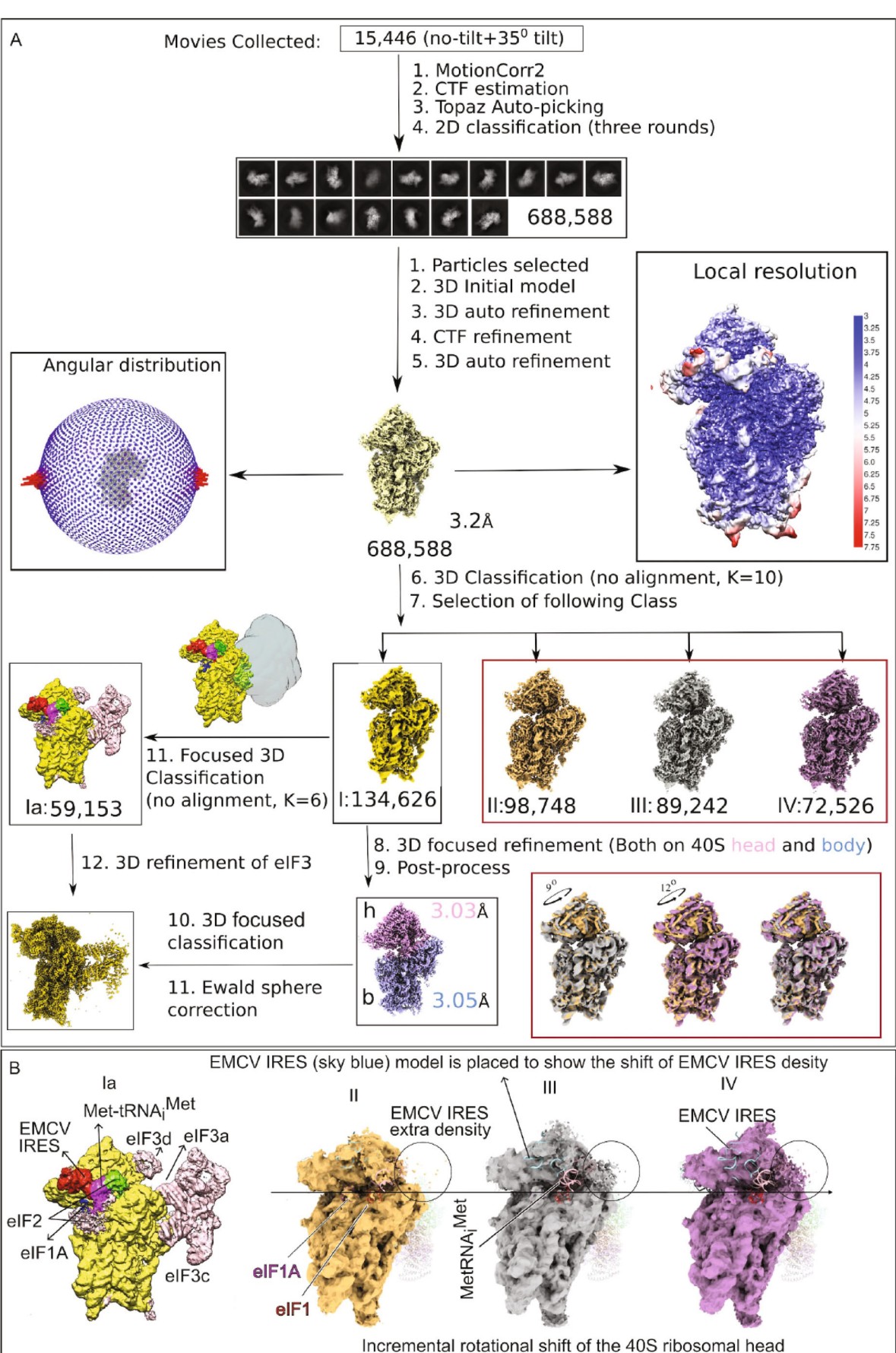

◀ **Figure EV3. Cryo-EM data processing workflow and structural characterization of EMCV IRES 48S initiation complexes.**

(**A**) Data processing pipeline for structure determination of EMCV IRES 48S initiation complexes. (**B**) Class Ia (left): 48S initiation complex showing Met-tRNA$_i^{Met}$, eIF2, eIF1A, eIF3 (eIF3a/c/d subunits are indicated), and EMCV IRES density bound at the 40S subunit head. Classes II–IV: A series of reconstructions capturing progressive 40S head rotation, accompanied by repositioning of the EMCV IRES. Fragmented extra density attributed to the IRES is highlighted (circled), and a sky-blue model of the EMCV IRES is overlaid to illustrate its positional shift. Gaussian filtering reveals that this density remains connected to the IRES across all states. The horizontal arrow from left to right denotes the direction of progressive rotational movement of the 40S head, representing a continuum of structural intermediates.

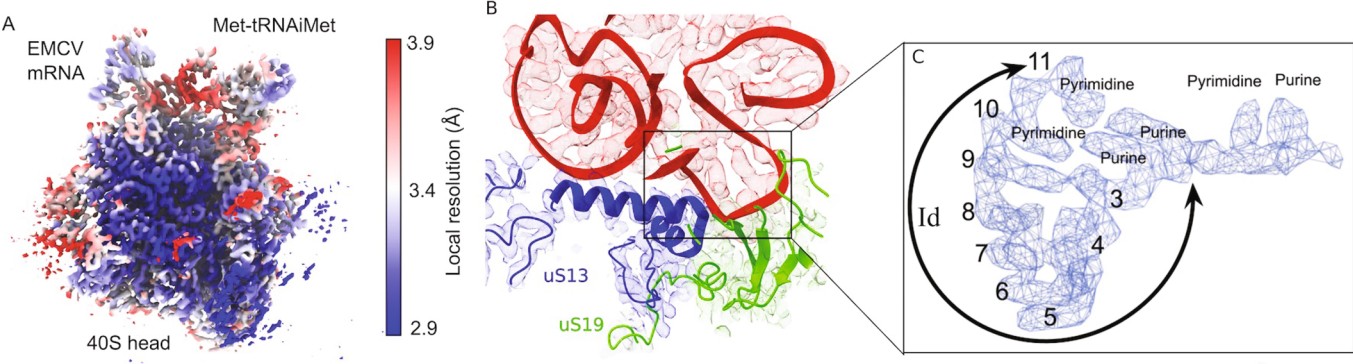

**Figure EV4. Local resolution estimation and assignment of nucleotides of EMCV mRNA.**

(A) The local resolution estimation of the 40S head part along with the EMCV IRES and Met-tRNA$_i^{Met}$. (B, C) The zoomed view of the Id subdomain of the EMCV IRES (in mesh) showing the loop formation through base pairing among RNA bases. The models fitted within densities of the EMCV IRES (red), uS19 (green), and uS13 (blue) are showing the relative position of Id on the 40S head.

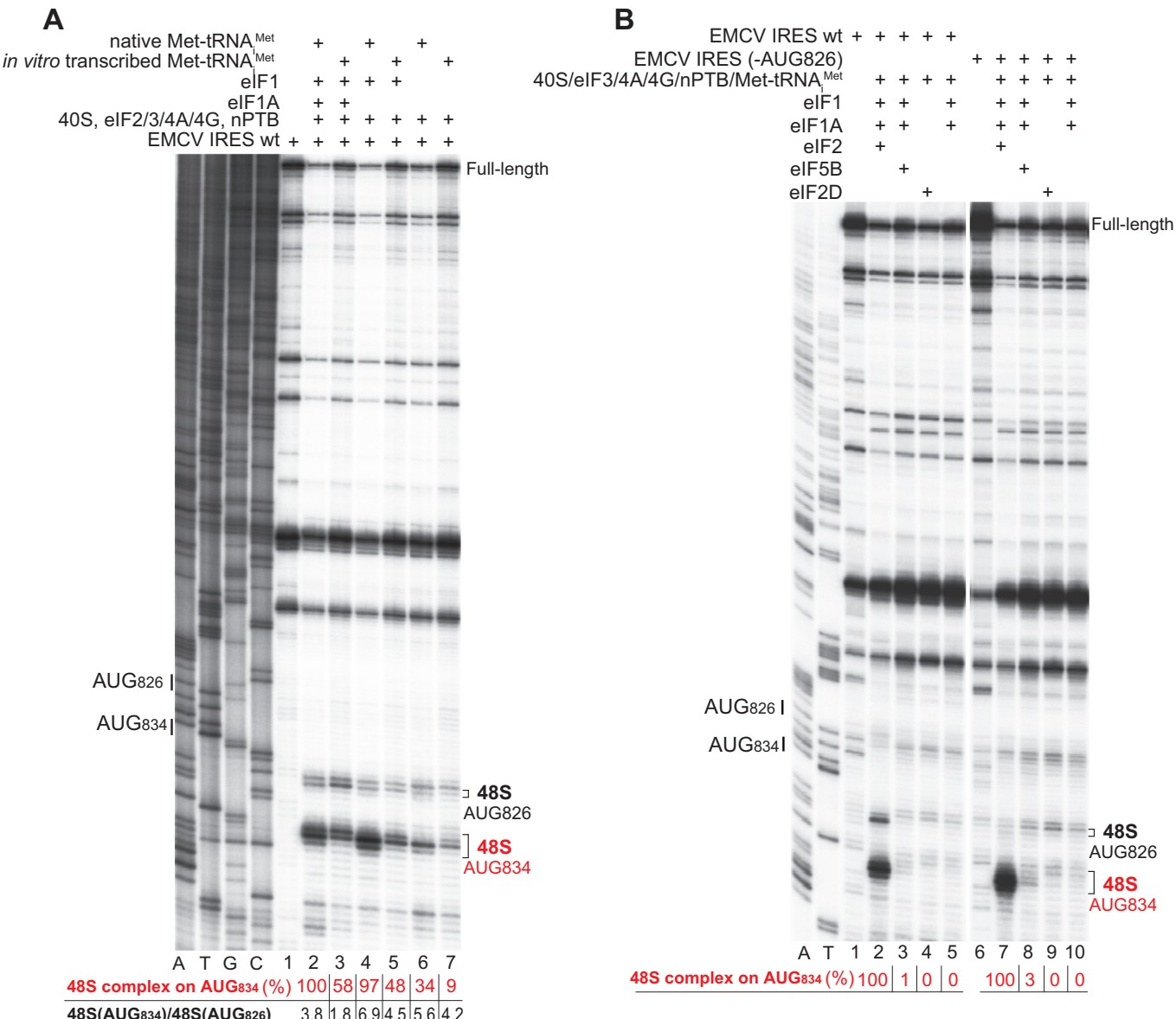

**Figure EV5.  Influence of the nature of initiator tRNA and substitution of eIF2 by eIF5B or eIF2D on 48S complex formation on the EMCV IRES.**

(A) Toe-printing analysis of 48S complex formation on wt EMCV IRES mRNA in the presence of either native or in vitro transcribed Met-tRNA$_i^{Met}$, 40S subunits, nPTB and indicated initiation factors. The efficiency of 48S complex formation on AUG$_{834}$ (red numbers) was quantified on the basis of three experiments with the efficiency of complex formation in the presence of eIF1, eIF1A and native Met-tRNA$_i^{Met}$ defined as 100%. Standard deviations (omitted for clarity) did not exceed 8%. Black numbers represent relative efficiencies of 48S complex formation on AUG$_{834}$ and AUG$_{826}$ in each condition. (B) Toe-printing analysis of 48S complex formation on wt EMCV IRES mRNA and mRNA with mutated AUG$_{826}$ in the presence of 40S subunits, initiator tRNA, nPTB and indicated initiation factors. The efficiency of 48S complex formation on AUG$_{834}$ (red numbers) was quantified taking the efficiency of complex formation in the presence of eIF2 defined as 100%. Standard deviations (omitted for clarity) did not exceed 15%. (A, B) Lanes C, T, A, and G depict wt EMCV sequence generated using the same primer. The positions of initiation codons are indicated on the left and of assembled 48S complexes on the right. The division between lanes 5 and 6 (B) indicates that these two sets of lanes were derived from the same gel. Source data are available online for this figure.

