## [Peer Review File · The EMBO Journal]

The mechanism of ribosomal recruitment during translation initiation on the Type 2 encephalomyocarditis virus IRES

Sayan Bhattacharjee, irina Abaeva, Zuben Brown, Yani Arhab, Hengameh Fallah, Christopher Hellen, Joachim Frank, and Tatyana Pestova

Corresponding author(s): Tatyana Pestova (tatyana.pestova@downstate.edu) , Christopher Hellen (christopher.hellen@downstate.edu), Joachim Frank (jf2192@cumc.columbia.edu)

Review Timeline:

Submission Date:	16th Jul 25
Editorial Decision:	15th Sep 25
Revision Received:	9th Nov 25
Editorial Decision:	11th Dec 25
Revision Received:	23rd Dec 25
Accepted:	2nd Feb 26

Editor: Cornelius Schneider

Transaction Report:

Dear Prof. Pestova,

Thank you for submitting your manuscript for consideration by the EMBO Journal. It has now been seen by three referees whose comments are shown below.

Given the referees' positive recommendations, I would like to invite you to submit a revised version of the manuscript, addressing the comments of all three reviewers. I should add that it is EMBO Journal policy to allow only a single round of revision, and acceptance of your manuscript will therefore depend on the completeness of your responses in this revised version. Please feel free to contact me if you have any questions regarding the revisions. I am also happy to discuss specific revision experiments via videoconferencing or e-mail.

Thank you for the opportunity to consider your work for publication. I look forward to your revision.

Yours sincerely,

Cornelius Schneider, PhD
Editor
The EMBO Journal
c.schneider@embojournal.org

Please remember: Digital image enhancement is acceptable practice, as long as it accurately represents the original data and conforms to community standards. If a figure has been subjected to significant electronic manipulation, this must be noted in the

figure legend or in the 'Materials and Methods' section. The editors reserve the right to request original versions of figures and the original images that were used to assemble the figure.

We realize that it is difficult to revise to a specific deadline. In the interest of protecting the conceptual advance provided by the work, we recommend a revision within 3 months (14th Dec 2025). Please discuss the revision progress ahead of this time with the editor if you require more time to complete the revisions. Use the link below to submit your revision:

Referee #1:

This manuscript reports the structure of picornavirus type 2 IRES RNA associated with the ribosome. Picornavirus IRES classified as type II, typically present in cardiovirus and aphthovirus, adopt a similar secondary structure, despite large differences in sequence. In the last years, the secondary structure of type II IRES, encompassing five domains (H-L), has been analyzed at high resolution. The long central domain, termed I, harbors conserved motifs at its apical region resembling a clover loop, including a GNRA tetraloop, which are essential for type II IRES activity. In addition, previous structural analysis showed the relevance of JK domains for association with eIF4G, although this domain alone is not sufficient to promote translation. Here, using in vitro reconstitution assays with purified components (eIFs, 40S, nPTB, and native Met-initiator tRNA), the authors have analyzed the assembly of 48S initiation complexes on the EMCV IRES by cryo-EM. This high-resolution structure (3.2Å) shows that the apical region of domain I directly contacts 40S ribosomal proteins uS13 and uS19. This interaction is further supported by the specific interaction of the conserved GNRA tetraloop with the Met-initiator tRNA. A detailed study of the influence of several nucleotides along domain I in 48S complexes recruitment by mutational analysis followed by footprint assays, RNA structure analysis (IRES wt and multiple mutants), and cryo-EM structure determination, is also shown. Furthermore, the manuscript shows that interaction of the IRES with the 40S subunit interferes with 60S subunit association, preventing formation of the 80S ribosomes. Authors suggest that IRES displacement disrupting the GNRA tetraloop/Met-tRNA interaction is required for ribosomal subunit joining, supported by the inability of eIF5B to substitute for eIF2 in 48S complex assembly. Based on the strong conservation of RNA structure, the authors claim that type II IRES domains shared a similar mechanism for 48S complex assembly, beyond eIF4G recognition. This is a well written breakthrough article which provides new molecular insights into the mechanism of translation initiation driven by robust viral IRESs.

Minor comments

Authors conclude that the mechanism of 48S recruitment by the EMCV IRES applies to all type II IRESs. It will be interesting to discuss whether this may also apply to type I IRESs, which also have a GNRA tetraloop in the equivalent stem-loop IV (PMID: 32556302, PMID: 15134450).

For the most part the literature is appropriately cited. However, previous data have analyzed the conformational flexibility of type II IRES by differential SHAPE (PMID: 29615727).

Referee #2:

This manuscript from Bhattacharjee et al. focuses on the biochemical mechanism of initiation from the EMCV IRES, the common example used by most for Type II IRESs. The work is a combination of new, more detailed cryo-EM and biochemical reconstitution of 48S initiation complexes with EMCV IRES mRNAs. In general, the manuscript is written more for the translational control field, especially those focused on ribosome/RNA structure and viral IRESs, compared to the broad audience of EMBO Journal. This is not really a negative point but it is noticeable, in my opinion. The new structural details are typically tested further with smart and careful mutations, which adds to the rigor of the work.

Suggestions to make the manuscript stronger:

1) The title states Type 2 IRESs, but only a single IRES was really tested. Yes, EMCV is the prototypical IRES for this class, but it could be over generalization.

2) The Results section is broken up into three major sections. The header/titles for these sections are not very informative and do not provide the major conclusion of the section. This results in a bit of an unorganized feel to the manuscript, especially for the third section with Figures 6 & 7.

3) The first paragraph of the Results section introduces what appears to be important data for the proper assembly of the 48S IRES initiation complex. But no data is actually shown. I understand the Pestova and Hellen labs have long established such techniques, but if the authors spend time walking through these data/concepts that are key, I recommend actually providing the

readers with these data, at least in the Supplemental Data.

4) Appendix Figure S2 is referenced and discussed before Appendix Figure S1.

5) Throughout the manuscript, the primer extension experiments (which are quite nice and rigorous) should be quantified which independent replicate datum plotted. I would argue the same for the hydroxy radical cleavage and RNase T1 digestion experiments. Having these data quantified with replicates would add rigor to the manuscript.

6) Any idea why eIF2beta was not resolved? Others have observed it in 48S initiation complexes.

7) On page 9, where the second results section starts, related to Figure 2, the authors write about the lh2 helix. I see lb and ld but not lh. I apologize if I missed it, but can the authors double check that this is labeled clearly? Also, the authors write about N103 of uS19, but I cannot find where this residue is pointed out in Figure 2G-L.

8) On page 9, with the sentence that starts with "Strikingly, the lc GNRA loop...", the authors cite Figure 3A-C, but this should probably be Figure 3A-E. Panel E really shows this better than A-D.

9) Related to Appendix Figure S7A, and I apologize if I overlooked them, but I cannot find the methods for isolating the native initiator tRNA.

10) Related to the results described at the end of page 10 and top of page 11, I'm confused on the conclusion of this section. There isn't really a clear model presented about the clashing and how it is related to eIF5B.

11) In the last results section, related to Figures 6 and 7, many mutants appear to unexpectedly have little effect on "IRES function". However, the authors only tested the first third or so of initiation. For example, 60S subunit joining and making elongation competent 80S was not tested. How do the authors know these steps are not affected and result in altered IRES-mediated translation?

12) Page 11 contains a one sentence paragraph. Can the authors double check this didn't belong with the paragraph before or after?

13) In the methods describing the in vitro reconstitution of 48S complexes (multiple places), the authors provide some detail about the reaction set up, including the volume of buffer A used. Is this volume the final volume of the reaction or the volume to what the other >8 components were added to? This is unclear but the final reaction volume is required for the readers to know the final concentration of the components.

Referee #3:

This study uses cryo-EM and primer extension assays/toeprinting to investigate EMCV Type 2 IRES recruitment to ribosomes during translation initiation. Purified components are used to form complexes and quite a large number of ribosome particles are observed, grouped in 4 classes, including a 48S population with EMCV, lacking eIF4A/G, which reveals interactions of parts of the IRES that haven't been thoroughly investigated. The authors reveal that domain I of the IRES contacts ribosomal proteins uS13/uS19 and that a conserved GNRA tetraloop engages with the 40S subunit and Met-tRNA_i. Mutational and functional analyses confirm these interactions are important for efficient initiation. Overall, the work provides solid structural and mechanistic evidence for an unknown aspect of how Type 2 IRESes subvert canonical cap-dependent initiation.

Significance: This work shifts focus from the well-established eIF4G/eIF4A-JK domain interaction to an upstream domain I "apex" module that directly contacts ribosomal proteins and tRNA_i, thereby extending the mechanistic picture of EMCV Type 2 IRES function beyond factor tethering to direct ribosome engagement.

This paper will be of interest to a broad audience, especially ribosome/translation researchers and virologists. The conclusions of the paper are justified based on the data, and the paper is nicely put together with straightforward findings, but some minor concerns should be addressed.

- The cryo-EM structure of EMCV-bound 48S complexes did not reveal the position of the eIF4G/4A-bound to the JK domain, leaving some uncertainty in the structural interpretation. To address this gap, it may be useful to incorporate an AlphaFold-predicted model of eIF4G/4A bound to the IRES. Such an addition could provide additional structural context to complement the cryo-EM data, thereby strengthening the overall conclusions and offering a clearer framework for future experimental validation.

- The authors should include an SDS-PAGE gel showing the protein components used in the cryo-EM grid preparation to demonstrate sample composition and purity.

- There is a recent eLife "reviewed preprint" by Das and Hussain (August 2025) that independently reports similar findings on EMCV IRES domain I interactions with uS13/uS19 and the GNRA tetraloop with initiator tRNA. The relationship to this work

should be discussed. The current work provides some improvements over that work, including experimental validation, mutation of the additional putative start site, etc. that could be discussed while acknowledging aspects in agreement between the studies.

- Pg 10, 2nd paragraph, deletion of two A residues having the biggest effect, residues should be residues
- Page 13: The statement: "... the interactions observed for the EMCV IRES must be common for all Type 2 IRESs." is stronger than the current evidence supports. While the data underscore the functional importance of this region, additional experimental validation with other Type 2 IRESs would be needed to establish the generality of this conclusion. And as the authors themselves note on pages 11-12, the upstream region of domain I remains structurally uncertain, with modeling studies suggesting pseudoknot formation in certain viruses. Framing this conclusion more cautiously would better align with the data presented.
- Page 15: The dichotomy presented between domain I and the JK domain in the statement: "The fact that the activity of the EMCV IRES was very tolerant to significant changes in the lower part of the central stem of domain I (disruption, stabilization, truncation or extension of various regions) (Fig. 6) suggests that its role might be merely as a connector between the two major functional elements". risks oversimplifying what is likely a dynamic and cooperative recruitment process. The observed tolerance for mutations in the central stem of domain I is interpreted as evidence for a "connector" role; however, alternative explanations (e.g., redundancy in RNA-protein interactions) should be considered.
- The first sentence of the discussion calls the study a breakthrough cryo-EM study. While the study is important and very nicely done, I think that wording is too strong for the discussion of most papers and should be toned down.
- The primers are currently listed within the running text. For improved readability and accessibility, it would be helpful if the authors could present the primer sequences in a dedicated table.

non-essential suggestions for improving the study:

- As the authors note, the sample was not cross-linked prior to cryo-EM grid preparation. Including a cross-linked sample or an eIF4A inhibitor such as a Rocaglate as was used in the 48S structures with eIF4F/eIF4A bound from the Ramakrishnan lab could be informative, as it may help stabilize and reveal the relative positioning of eIF4G and eIF4A. Additionally, cross-linking could help with preferred orientation and improve particle distribution, potentially enhancing the overall quality of the cryo-EM data.

Referee #1:

This manuscript reports the structure of picornavirus type 2 IRES RNA associated with the ribosome. Picornavirus IRES classified as type II, typically present in cardiovirus and aphthovirus, adopt a similar secondary structure, despite large differences in sequence. In the last years, the secondary structure of type II IRES, encompassing five domains (H-L), has been analyzed at high resolution. The long central domain, termed I, harbors conserved motifs at its apical region resembling a clover loop, including a GNRA tetraloop, which are essential for type II IRES activity. In addition, previous structural analysis showed the relevance of JK domains for association with eIF4G, although this domain alone is not sufficient to promote translation. Here, using in vitro reconstitution assays with purified components (eIFs, 40S, nPTB, and native Met-initiator tRNA), the authors have analyzed the assembly of 48S initiation complexes on the EMCV IRES by cryo-EM. This high-resolution structure (3.2Å) shows that the apical region of domain I directly contacts 40S ribosomal proteins uS13 and uS19. This interaction is further supported by the specific interaction of the conserved GNRA tetraloop with the Met-initiator tRNA. A detailed study of the influence of several nucleotides along domain I in 48S complexes recruitment by mutational analysis followed by footprint assays, RNA structure analysis (IRES wt and multiple mutants), and cryo-EM structure determination, is also shown. Furthermore, the manuscript shows that interaction of the IRES with the 40S subunit interferes with 60S subunit association, preventing formation of the 80S ribosomes. Authors suggest that IRES displacement disrupting the GNRA tetraloop/Met-tRNA interaction is required for ribosomal subunit joining, supported by the inability of eIF5B to substitute for eIF2 in 48S complex assembly. Based on the strong conservation of RNA structure, the authors claim that type II IRES domains shared a similar mechanism for 48S complex assembly, beyond eIF4G recognition. This is a well written breakthrough article which provides new molecular insights into the mechanism of translation initiation driven by robust viral IRESs.

We are very grateful to the reviewer for the appreciative comments concerning our manuscript. They are not only flattering for the PIs, but also very encouraging for the postdocs who conducted the experiments.

Minor comments

Authors conclude that the mechanism of 48S recruitment by the EMCV IRES applies to all type II IRESs. It will be interesting to discuss whether this may also apply to type I IRESs, which also have a GNRA tetraloop in the equivalent stem-loop IV (PMID: 32556302, PMID: 15134450). For the most part the literature is appropriately cited. However, previous data have analyzed the conformational flexibility of type II IRES by differential SHAPE (PMID: 29615727).

Response: We are very grateful to the reviewer for pointing out that we overlooked the study by Lozano et al., 2018 (PMID: 29615727). We corrected this mistake and now refer to it in the Introduction of the revised manuscript. Regarding a discussion of a potential role of the GNRA tetraloop in Type 1 IRESs, we are addressing this in a separate study which will be submitted shortly and will therefore not discuss it here.

Referee #2:

This manuscript from Bhattacharjee et al. focuses on the biochemical mechanism of initiation from the EMCV IRES, the common example used by most for Type II IRESs. The work is a combination of new, more detailed cryo-EM and biochemical reconstitution of 48S initiation complexes with EMCV IRES mRNAs. In general, the manuscript is written more for the translational control field, especially those focused on ribosome/RNA structure and viral IRESs, compared to the broad audience of EMBO Journal. This is not really a negative point but it is noticeable, in my opinion. The new structural details are typically tested further with smart and careful mutations, which adds to the rigor of the work.

We thank the reviewer for appreciation of our work, very careful reading of the manuscript and for making valuable suggestions for its improvement.

Suggestions to make the manuscript stronger:

1) The title states Type 2 IRESs, but only a single IRES was really tested. Yes, EMCV is the prototypical IRES for this class, but it could be over generalization.

Response: The title of the manuscript has been revised accordingly.

2) The Results section is broken up into three major sections. The header/titles for these sections are not very informative and do not provide the major conclusion of the section. This results in a bit of an unorganized feel to the manuscript, especially for the third section with Figures 6 & 7.

Response: The subtitles of the Result sections have been modified.

3) The first paragraph of the Results section introduces what appears to be important data for the proper assembly of the 48S IRES initiation complex. But no data is actually shown. I understand the Pestova and Hellen labs have long established such techniques, but if the authors spend time walking through these data/concepts that are key, I recommend actually providing the readers with these data, at least in the Supplemental Data.

Response: The requested Figure showing toe-printing analysis of 48S complex formation on the mutant EMCV IRES used for this cryo-EM study is now shown as the new Expanded View Figure EV2B of the revised manuscript.

4) Appendix Figure S2 is referenced and discussed before Appendix Figure S1.

Response: This is not correct regarding referencing – Appendix Figure S1 was first mentioned in the Introduction, and following the journal's rules, it was therefore referenced before Appendix Figure S2. In the revised manuscript, Appendix Figure S1 is presented as Expanded View Figure EV1.

5) Throughout the manuscript, the primer extension experiments (which are quite nice and rigorous) should be quantified which independent replicate datum plotted. I would argue the same for the hydroxy radical cleavage and RNase T1 digestion experiments. Having these data quantified with replicates would add rigor to the manuscript.

Response: Toe-printing and RNase T1 digestion experiments have been quantified as requested. However, for the reason of space, the resulted numbers are not plotted but are shown underneath

the corresponding lane of the gels, as we did previously (e.g. PMID: 34883515). We did not quantify the directed hydroxyl radical cleavage experiments because cleavage occurred at a single position (CCA₅₇₃₋₅₇₅) so that there is no reference for comparison.

6) Any idea why eIF2beta was not resolved? Others have observed it in 48S initiation complexes.

Response: A plausible explanation for the unresolved density of eIF2β is the inherently flexible nature of the structure at this site. This intrinsic flexibility and resulting mobility is likely further amplified by the disruption of the stable architecture at its binding sites on the 40S subunit upon EMCV IRES binding, resulting in displacement of the tRNA acceptor arm and rotation of the 40S head and destabilization of the interaction network. Resolving a stable complex by cryo-EM requires a well-defined and rigid structural arrangement, and parts of the structure that are mobile will show up with blurred or scattered density. Some researchers use cross-linking to artificially stabilize such parts. In this way visualization of eIF2β in the 48S initiation complex was achieved (<https://www.nature.com/articles/s41594-024-01378-4>, <https://www.science.org/doi/10.1126/science.aba4904>). We are committed to studying 48S complexes in their native states and have not resorted to this practice. The absence of well-resolved density for eIF2β is now discussed in the revised manuscript (page 10, lines 19-22; new Appendix Figure S4).

7) On page 9, where the second results section starts, related to Figure 2, the authors write about the Ih2 helix. I see Ib and Id but not Ih. I apologize if I missed it, but can the authors double check that this is labeled clearly? Also, the authors write about N103 of uS19, but I cannot find where this residue is pointed out in Figure 2G-L.

Response: Labeling of Figure 2 has been corrected.

8) On page 9, with the sentence that starts with "Strikingly, the Ic GNRA loop...", the authors cite Figure 3A-C, but this should probably be Figure 3A-E. Panel E really shows this better than A-D.

Response: The reviewer is correct; indeed, we even consider that it is better to expand the reference to Figure 3A-G, as done in the revised manuscript.

9) Related to Appendix Figure S7A, and I apologize if I overlooked them, but I cannot find the methods for isolating the native initiator tRNA.

Response: The reviewer did not miss anything. We did not isolate native initiator tRNA. Instead, we aminoacylated native total calf liver tRNA (Promega) using recombinant *E. coli* methionyl-tRNA synthetase that aminoacylates only initiator but not elongator tRNA^{Met}, which allows exact quantification of aminoacylation. Aminoacylation of native and *in vitro* transcribed tRNAs is described in Materials and Methods.

10) Related to the results described at the end of page 10 and top of page 11, I'm confused on the conclusion of this section. There isn't really a clear model presented about the clashing and how it is related to eIF5B.

Response: We apologize for creating the confusion. We now expanded and clarified the corresponding text and also introduced some changes to the corresponding Figure (page 11, line 25 to page 12, line 16; Figure 5 of the revised manuscript).

11) In the last results section, related to Figures 6 and 7, many mutants appear to unexpectedly have little effect on "IRES function". However, the authors only tested the first third or so of initiation. For example, 60S subunit joining and making elongation competent 80S was not tested. How do the authors know these steps are not affected and result in altered IRES-mediated translation?

Response: The reviewer is absolutely correct that we only tested the influence of these mutations on 48S complex formation. Therefore, we now state in the Discussion that the influence of these regions of the IRES on ribosomal subunit joining cannot be excluded (page 16, lines 14-15 of the revised manuscript).

12) Page 11 contains a one sentence paragraph. Can the authors double check this didn't belong with the paragraph before or after?

Response: This sentence describes mutations in a separate distinct region of the IRES. For this reason, it must stay as a separate paragraph.

13) In the methods describing the in vitro reconstitution of 48S complexes (multiple places), the authors provide some detail about the reaction set up, including the volume of buffer A used. Is this volume the final volume of the reaction or the volume to what the other >8 components were added to? This is unclear but the final reaction volume is required for the readers to know the final concentration of the components.

Response: As requested, the final volumes of reaction mixtures are now clearly stated throughout Materials and Methods of the revised manuscript.

Referee #3:

This study uses cryo-EM and primer extension assays/toeprinting to investigate EMCV Type 2 IRES recruitment to ribosomes during translation initiation. Purified components are used to form complexes and quite a large number of ribosome particles are observed, grouped in 4 classes, including a 48S population with EMCV, lacking eIF4A/G, which reveals interactions of parts of the IRES that haven't been thoroughly investigated. The authors reveal that domain I of the IRES contacts ribosomal proteins uS13/uS19 and that a conserved GNRA tetraloop engages with the 40S subunit and Met-tRNA_i. Mutational and functional analyses confirm these interactions are important for efficient initiation. Overall, the work provides solid structural and mechanistic evidence for an unknown aspect of how Type 2 IRESes subvert canonical cap-dependent initiation.

Significance: This work shift focuses from the well-established eIF4G/eIF4A-JK domain interaction to an upstream domain I "apex" module that directly contacts ribosomal proteins and tRNA_i, thereby extending the mechanistic picture of EMCV Type 2 IRES function beyond factor tethering to direct ribosome engagement.

This paper will be of interest to a broad audience, especially ribosome/translation researchers and virologists. The conclusions of the paper are justified based on the data, and the paper is nicely put together with straightforward findings, but some minor concerns should be addressed.

We thank the reviewer for the appreciative comments concerning our manuscript and for the very helpful suggestions.

Minor concerns:

- The cryo-EM structure of EMCV-bound 48S complexes did not reveal the position of the eIF4G/4A-bound to the JK domain, leaving some uncertainty in the structural interpretation. To address this gap, it may be useful to incorporate an AlphaFold-predicted model of eIF4G/4A bound to the IRES. Such an addition could provide additional structural context to complement the cryo-EM data, thereby strengthening the overall conclusions and offering a clearer framework for future experimental validation.

Response: A model of the structural organization of the EMCV IRES 48S initiation complex can be built by aligning the existing structures of eIF4G/4A-interacting JK domain (PDB ID: 8J7R), of eIF4G/4A-containing 48S complexes (PDB ID: 8OZ0), and of the cryo-EM structure of 48S complexes assembled on the EMCV IRES (this study) (see below). However, we don't want to include this model because, as we stated in the last paragraph of the Discussion, initiation on the EMCV IRES does not require the portion of eIF4G that is responsible for its interaction with eIF3, which poses the questions of whether the JK domain plays the role in positioning of the eIF4G/eIF4A/JK domain complex by specific interactions with e.g. the 40S subunit or eIF3 and whether the position of eIF4G/eIF4A in initiation complexes assembled on the EMCV IRES is the same as in canonical initiation complexes or if the JK domain usurps eIF4G/eIF4A and displaces them from their canonical binding site. Thus, we don't want to present a model which could be found to be wrong in the near future.

Figure: Model of structural organization of the EMCV IRES 48S initiation complex. (A) Overview of the complex with eIF2 α and various domains. (B) A 180° rotated view highlighting key interactions. (C) 90° rotated view showing missing density of the rest of the EMCV IRES with red dotted line. (D) Detailed view of the EMCV IRES J-K domains interacting with eIF4G and eIF4A. The model was generated by aligning the eIF4G domain from PDB ID: 8J7R with the eIF4G domain in PDB ID: 8OZO

- The authors should include an SDS-PAGE gel showing the protein components used in the cryo-EM grid preparation to demonstrate sample composition and purity.

Response: The requested SDS-PAGE gel is included in the revised manuscript as the new Expanded View Figure EV2A.

- There is a recent eLife "reviewed preprint" by Das and Hussain (August 2025) that independently reports similar findings on EMCV IRES domain I interactions with uS13/uS19 and the GNRA tetraloop with initiator tRNA. The relationship to this work should be discussed. The current work provides some improvements over that work, including experimental validation, mutation of the additional putative start site, etc. that could be discussed while acknowledging aspects in agreement between the studies.

Response: As requested, we acknowledged similar interactions between the EMCV IRES and components of the 48S initiation complex that were observed in the recent eLife "reviewed preprint" by Das and Hussain (August 2025) (page 14, lines 4-6 of the revised manuscript). However, a more detailed comparison is not possible because that structure is of lower resolution and the report also lacks functional data.

- Pg 10, 2nd paragraph, deletion of two A residues having the biggest effect, residues should be residues

Response: This typographical error has been corrected.

- Page 13: The statement: "... the interactions observed for the EMCV IRES must be common for all Type 2 IRESs." is stronger than the current evidence supports. While the data underscore the functional importance of this region, additional experimental validation with other Type 2 IRESs would be needed to establish the generality of this conclusion. And as the authors themselves note on pages 11-12, the upstream region of domain I remains structurally uncertain, with modeling studies suggesting pseudoknot formation in certain viruses. Framing this conclusion more cautiously would better align with the data presented.

Response: Although we do believe that the observed interactions are common for all type 2 IRESs, this statement was nevertheless toned down (page 14, line 12 of the revised manuscript).

- Page 15: The dichotomy presented between domain I and the JK domain in the statement: "The fact that the activity of the EMCV IRES was very tolerant to significant changes in the lower part of the central stem of domain I (disruption, stabilization, truncation or extension of various regions) (Fig. 6) suggests that its role might be merely as a connector between the two major functional elements". risks oversimplifying what is likely a dynamic and cooperative recruitment process. The observed tolerance for mutations in the central stem of domain I is interpreted as evidence for a "connector" role; however, alternative explanations (e.g., redundancy in RNA-protein interactions) should be considered.

Response: In agreement with the reviewer, we expanded discussion of the possible roles of the lower part of the central stem of domain I (page 16, lines 14-15 of the revised manuscript).

- The first sentence of the discussion calls the study a breakthrough cryo-EM study. While the study is important and very nicely done, I think that wording is too strong for the discussion of most papers and should be toned down.

Response: We do believe that this structure represents a breakthrough in studying of the mechanism of IRES-mediated translation and reviewer #1 seemed to agree with that. The EMCV IRES was discovered in 1988 (PMID: 2839690), is considered to be a paradigm for end-independent initiation of translation in eukaryotes and is widely used in biotech applications. However, although the minimum set of factors required for 48S complex formation on this IRES was determined in 1996 (PMID: 8943341, 8943342), the mechanism for ribosomal recruitment to this IRES has remained obscure, particularly since the discovery that it did not depend on interaction of eIF4G in the eIF4G/eIF4A/IRES complex with eIF3 in the 43S complex (PMID: 10913184). Our determination of essential interactions of the IRES with the 40S subunit and, completely unexpectedly, with initiator tRNA constitutes a major advance, so that we considered that the use of "breakthrough" is justified.

- The primers are currently listed within the running text. For improved readability and accessibility, it would be helpful if the authors could present the primer sequences in a dedicated table.

Response: As requested, the primer sequences are now presented in the new dedicated Appendix Table S4.

non-essential suggestions for improving the study:

- As the authors note, the sample was not cross-linked prior to cryo-EM grid preparation. Including a cross-linked sample or an eIF4A inhibitor such as Rocaglate as was used in the 48S

structures with eIF4F/eIF4A bound from the Ramakrishnan lab could be informative, as it may help stabilize and reveal the relative positioning of eIF4G and eIF4A. Additionally, cross-linking could help with preferred orientation and improve particle distribution, potentially enhancing the overall quality of the cryo-EM data.

Response: We thank the reviewer for this valuable suggestion. In future, we do plan to attempt cryo-EM studies using cross-linked Type 2 IRES-containing ribosomal complexes with the aim of revealing the position of the eIF4G/4A-bound JK domain.

Dear Prof. Pestova,

Thank you for submitting a revised version of your manuscript. Your study has now been seen by 2 of the original referees. As you can see from the reports referee #2 reiterates that the study would benefit from showing and quantifying replicates. We agree with the referee that showing replicates for all experiments is important would therefore ask you to provide these as supplementary figures. Other than that, both referees find their previous concerns have been addressed and now recommend publication of the manuscript. In addition to the request for replicates there remain only a few mainly editorial points that have to be addressed before I can extend formal acceptance of the manuscript:

- 1) AFFILIATIONS (research institution or university vs. biotech company): employment in a biotech company should be stated in "disclosure of competing interest"
- 2) Please place the keywords below the Abstract
- 3) Please remove the DOI numbers in each reference
- 4) Please rename the Conflict of Interest section into "Disclosure and Competing Interests Statement", in accordance with our updated Guide to Authors (<https://link.springer.com/partners/embo-press/editorial-policies#Competing%20interest%20disclosures>)
- 5) As we are switching from a free-text author contribution statement towards a more formal statement based on Contributor Role Taxonomy (CRediT) terms, please remove the present Author Contribution section and instead specify each author's contribution(s) directly in the Author Information page of our submission system during upload of the final manuscript. See <https://casrai.org/credit/> for more information.
- 6) APPENDIX 1 FILE WITH ToC: ok, but PDB validation report should be removed from Appendix PDF and uploaded as an individual file (Related Manuscript File)
- 7) Thank you for providing the SOURCE DATA including the complete checklist, but update either zip folder or SD checklist as there is SD for Fig. 4, but stated Fig. 3 in SD checklist
- 8) Please provide an altered synopsis image to accompany the already provided synopsis blueb, making sure that the aspect ratio conforms to our website's format - it should be exactly 550 pixels wide and between 300-600 pixels high.
- 9) Please provide the specific URLs for EMD-40769, EMD-40770, EMD-40771, EMD-40772, EMD-40773, EMD-40774, 8SUP datasets in the data availability statement.
- 10) Please rename the "Materials and Methods" section to "Methods"
- 11) Sections need to be named and the order should be corrected: Title page - Abstract - Keywords - Introduction - Results - Discussion - Methods - Data Availability - Acknowledgements - Disclosure and Competing Interests Statement - References - Figure Legends - Table(s) - Expanded View Figure Legends.

With best regards,
Cornelius Schneider

Cornelius Schneider, PhD
Editor | The EMBO Journal
c.schneider@embojournal.org

Please refer to our figure preparation guideline in order to ensure proper formatting and readability in print as well as on screen:

<https://link.springer.com/journal/44318/submission-guidelines#cms-Figure-and-data-presentation>

Use the link below to submit your revision:

Referee #1:

The manuscript can be accepted as it is

Referee #2:

The authors have done a good job at addressing the concerns. While I can understand the choice of quantitating the single gel presented for each experiment due to space issues, the authors can easily display replicate gels and, at the very least, plots with each data point with standard deviation and statistical comparisons in the Supplementary Material. The authors state the SD was not included for clarity; I would argue that presenting the experimental error within data provides the readers the clarity and transparency that is required and now standard & expected in the field. Without such data, the conclusions are assumed to be from an $n=1$, which is not acceptable in the field.

All minor editorial requests have been addressed by the authors.

Dear Prof. Pestova,

I am pleased to inform you that your manuscript has been accepted for publication in the EMBO Journal.

You may qualify for financial assistance for your publication charges - either via a Springer Nature fully open access agreement or an EMBO initiative. Check your eligibility: <https://link.springer.com/journal/44318/how-to-publish-with-us>

Yours sincerely,

Cornelius Schneider, PhD
Editor
The EMBO Journal
c.schneider@embojournal.org

Please note that it is The EMBO Journal policy for the transcript of the editorial process (containing referee reports and your response letters) to be published as an online supplement to each paper. If you should prefer removal of any referee-only figures included in the point-by-point response(s), e.g. because they may still be used for future publication or because they have been reproduced from published work by others, please do let us know immediately via response email.

More information is available here: <https://link.springer.com/partners/embo-press/editorial-policies#Peer%20review>